# DyFlow: Dynamic Workflow Framework for Agentic Reasoning

**Yanbo Wang**[1], **Zixiang Xu**[1], **Yue Huang**[2], **Xiangqi Wang**[2], **Zirui Song**[1]
**Lang Gao**[1], **Chenxi Wang**[1], **Xiangru Tang**[3], **Yue Zhao**[4], **Arman Cohan**[3]
**Xiangliang Zhang**[2], **Xiuying Chen**[1,†]

[1]Mohamed bin Zayed University of Artificial Intelligence (MBZUAI)
[2]University of Notre Dame, [3]Yale University, [4]University of Southern California

Email: `wyf23187@gmail.com`    [†]Corresponding author

## Abstract

Agent systems based on large language models (LLMs) have shown great potential in complex reasoning tasks, but building efficient and generalizable workflows remains a major challenge. Most existing approaches rely on manually designed processes, which limits their adaptability across different tasks. While a few methods attempt automated workflow generation, they are often tied to specific datasets or query types and make limited use of intermediate feedback, reducing system robustness and reasoning depth. Moreover, their operations are typically predefined and inflexible. To address these limitations, we propose **DyFlow**, a dynamic workflow generation framework that adaptively constructs and adjusts reasoning procedures based on task requirements and real-time intermediate feedback, thereby enhancing cross-task generalization. DyFlow consists of two core components: a designer and an executor. The designer decomposes complex problems into a sequence of sub-goals defined by high-level objectives and dynamically plans the next steps based on intermediate outputs and feedback. These plans are then carried out by the executor, which executes each operation using dynamic operators with context-aware parameterization, enabling flexible and semantically grounded reasoning. We systematically evaluate DyFlow across diverse domains, including social reasoning, biomedical tasks, mathematical problem solving, and code generation. Results demonstrate that DyFlow significantly outperforms existing baselines, achieving substantial Pass@k improvements and exhibiting robust generalization across diverse domains. The code is publicly available at `https://github.com/wyf23187/DyFlow`.

## 1 Introduction

Large language models (LLMs) have demonstrated remarkable abilities in language understanding, generation, and complex reasoning tasks [1, 2, 3]. They support many applications, from dialogue systems and content generation to autonomous agents and multi-step decision-making [4, 5, 6, 7]. Recently, there has been a growing trend of deploying LLMs as *agents*, where multiple LLMs collaborate to tackle complex tasks. Debate-based systems enable agents to critique each other's solutions [8], and team-based agents distribute tasks across specialized roles [9, 10].

However, most existing multi-agent frameworks use *static*, pre-defined workflows, as illustrated in Figure 1. Each agent's role and task sequence is fixed beforehand, proceeding rigidly without intermediate feedback. For instance, CAMEL [10] assigns agents pre-defined roles via system prompts, MetaGPT [9] enforces collaboration based on standard operating procedures, and AutoGen

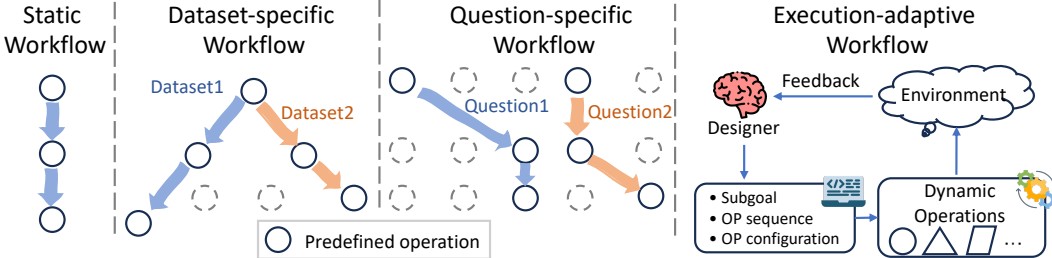

Figure 1: Paradigm shift in LLM-based reasoning workflows. From left to right: static workflows apply fixed sequences across all tasks; dataset-specific and question-specific workflows allow more variation but rely on predefined operations (OP); DyFlow adopts an execution-adaptive paradigm that dynamically adjusts the workflow based on intermediate feedback.

[5] and OpenAgents [11] orchestrate agents through static communication graphs or APIs. When a sub-task fails or produces flawed output, these systems typically halt or propagate errors due to the lack of mechanisms for revising plans or goals. Recent frameworks introduce some adaptability but with limitations: AFlow [12] and ADAS [13] optimize workflows via offline search guided by dataset-level performance, still tied to specific training distributions. DyLAN [14] and MaAS [15] offer query-specific agent configurations but do not adjust subgoals based on real-time feedback.

To address these limitations, we propose **DyFlow**, a feedback-driven framework for LLM-based agents that dynamically adapts subgoal planning during execution. DyFlow is built on a hierarchical *designer-executor* architecture and operates in a sequenced manner: Given a task, the high-level *designer* initializes the process by planning the first subgoal and its corresponding operator execution graph based on the task requirements. The low-level *executor* then carries out this subgoal, leveraging tools, APIs, or external functions as required. At each execution step, DyFlow collects intermediate outputs, tool feedback, and possible error signals. Based on this updated context, the designer generates a revised subgoal plan tailored to the current state. This feedback loop continues throughout the task, enabling DyFlow to adjust its reasoning trajectory in real time. If a subgoal fails or encounters unexpected results, DyFlow can proactively revise the plan, retry the subgoal, or reassign responsibilities—supporting both high-level strategy shifts and fine-grained execution corrections. Compared to prior methods such as DyLAN [14] and ReAct [16], which only adapt agent roles or individual actions, DyFlow enables dynamic restructuring at the subgoal level, offering greater robustness and flexibility in complex, real-world tasks.

While the executor does not require any additional training and can be directly instantiated using existing open-source or proprietary LLMs, the designer is trained separately to acquire strong planning capabilities. To this end, we employ a two-phase learning strategy to empower lightweight models to function as effective designers comparable to large proprietary models. In the initialization phase, the designer is trained via supervised fine-tuning on a collection of successful planning examples, allowing it to acquire basic planning capabilities. This is followed by a self-play phase, where the designer interacts with the executor to generate its own execution trajectories and iteratively refine its planning policy based on feedback. The resulting feedback trains the designer to favor successful plans and learn from failures. Unlike static expert demonstrations, this method internalizes the dynamic feedback loop for adaptive planning. Our approach does not impose additional training requirements on the executor, enabling direct use of existing open-source or proprietary LLMs.

We evaluate DyFlow across diverse reasoning domains, including causal, math, code, medical, and social reasoning. Experimental results demonstrate that DyFlow significantly outperforms strong baselines. Notably, it exhibits robust cross-domain generalization and adaptability to different executor models, even when deployed with unseen LLMs.

Our contributions are summarized as follows:

- We introduce **DyFlow**, a dynamic planning framework for LLM-based agents that adaptively revises reasoning procedures based on real-time feedback. DyFlow consists of two key components: a high-level *designer* that generates and revises subgoal plans, and a low-level *executor* that carries out the subgoals.
- We develop a designer training method that equips lightweight models with strong structured reasoning ability, achieving performance comparable to proprietary LLMs.

- We demonstrate DyFlow's robustness across diverse domains, models, and tasks through systematic experiments covering generalization, efficiency, and ablation.

## 2 Related Work

### 2.1 Static and Dynamic Agentic Workflows

Many LLM-based agent systems are built upon static workflows, where agent roles, execution order, and interaction protocols are predefined and fixed throughout the task. For example, MetaGPT [9] follows standard operating procedures (SOPs) encoded into prompt templates to coordinate collaborative coding, while AutoGen [5] orchestrates multi-agent dialogues through rigid communication graphs. CAMEL [10] and OpenAgents [11] further reinforce this paradigm by hard-coding agent roles or API usage. Although such designs facilitate structured cooperation, they lack the capacity to revise plans in response to failures or evolving context, resulting in limited robustness and adaptability.

To reduce reliance on manual design, a line of research has explored automated agent construction. AFlow [12] refines code-centric workflows using Monte Carlo Tree Search, ADAS [13] performs meta-search over agent architectures, and AgentSquare [17] evolves modular components via composition. These systems automate workflow design, yet they still adopt one-shot planning: once constructed, the workflow remains fixed during execution. Other methods such as DyLAN [14] and MaAS [15] dynamically configure agents or select templates based on task complexity, but lack mechanisms to adapt subgoal plans based on runtime feedback. More recently, ScoreFlow [18] and MaAS introduce continuous or conditional search spaces over workflows and agent architectures, offering more expressive design flexibility. However, they still treat planning as a pre-execution optimization problem, separated from the execution process itself.

### 2.2 Feedback-based Correction in LLM Systems

Intermediate feedback has been explored for improving LLM reasoning ability across various settings. In single-agent systems, Reflexion [19] performs retrospective self-evaluation across episodes, while Tree of Thoughts [20] enables trajectory-level revision via search. DSPy [21] introduces assertion-based checks to detect and fix failures at the module level. In multi-agent frameworks, AutoGen [5] supports retrying failed actions within fixed communication rounds.

However, these correction mechanisms have been underutilized in workflow planning. Effective correction at the workflow level requires more than recovering from execution failures, as it demands the ability to dynamically revise subgoal decomposition and operator selection in response to intermediate signals. This poses stricter requirements for granularity and adaptability than existing designs can satisfy. DyFlow addresses this gap by directly integrating feedback into the planning loop. It adjusts subgoal plans and operator configurations in real time based on contextual signals, enabling fine-grained and resilient reasoning workflows that evolve with task progress.

## 3 Methodology: DyFlow Framework

In this section, we introduce **DyFlow**, a reasoning framework that constructs dynamic workflows through iterative subgoal planning and feedback integration. At each step, a designer generates a task-specific plan, represented as a stage subgraph, which is conditioned on the current context and intermediate outputs. This design enables DyFlow to adapt its reasoning trajectory in response to execution results, offering greater flexibility than static or template-based workflows.

### 3.1 Formalization of DyFlow Planning

To support our dynamic workflow construction, this subsection formalizes DyFlow's planning mechanism, detailing its state representation, operator templates, and the structure of stage subgraphs.

DyFlow formulates complex reasoning tasks as a dynamic sequence of decision-making steps. Each task $p \in \mathcal{P}$ requires a series of interdependent actions to reach a solution. At every step $t$, the system maintains a state $s_t$, which captures the complete context necessary for planning and execution. $s_t$ includes the original task specification, previously generated plans $(G_0, \ldots, G_{t-1})$, intermediate

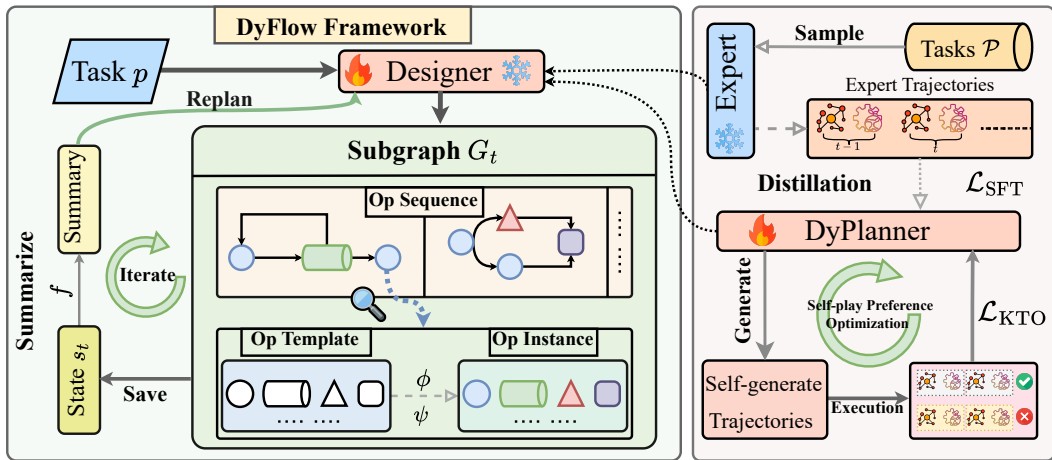

Figure 2: DyFlow dynamically constructs reasoning workflows by generating stage subgraphs based on the current task state. A high-level designer plans operator sequences, while a low-level executor executes them using memory. The designer is trained via supervised distillation and self-play preference optimization.

outputs such as partial answers or review verdicts, and any encountered errors. The process begins with an initial state $s_0$, containing only the task itself, and this state is progressively updated as the system executes each planning step.

DyFlow draws from a finite set of **operator templates** $\mathcal{O} = \{O_1, \ldots, O_N\}$, each corresponding to a basic type of reasoning or execution step. These templates define reusable operations that can be instantiated according to context. At each step, DyFlow constructs a structured plan represented as a stage subgraph $G_t = (V_t, E_t, v_{\text{start}}^t, C_{\text{end}}^t)$. This directed graph specifies a set of actions to execute: $V_t$ contains operator instances, each instantiated from a template in $\mathcal{O}$. Formally, **an operator instance** $o \in V_t$ is defined as a tuple $o = (O_k, \phi, \psi)$, where $O_k \in \mathcal{O}$ is the template, $\phi$ is a fine-grained instruction tailored to the current context, and $\psi$ is a list of input keys referencing a global memory buffer $\mathcal{M}$, which stores previous outputs as key–value pairs. This design allows operator instances to flexibly reuse information across reasoning steps. The edge set $E_t$ encodes dependencies between these operations, $v_{\text{start}}^t$ marks the entry point of execution, and $C_{\text{end}}^t$ defines conditions for plan termination. A complete list of operator templates and their descriptions is provided in Appendix C.

A full task execution generates a trajectory $\tau = (s_0, G_0, s_1, G_1, \ldots, G_{T-1}, s_T)$, where each state transition $s_t \to s_{t+1}$ results from executing the corresponding plan $G_t$. Execution halts when the system triggers a terminate action, reaches a predefined step limit $T_{\max}$, or encounters an unrecoverable error. Through this formulation, DyFlow supports fine-grained, context-aware reasoning that can adaptively refine its strategy as execution unfolds. A formal theoretical analysis of DyFlow's performance under dynamic planning is included in Appendix B.

## 3.2 Planning and Execution Process

Building on the formal definition of DyFlow's planning mechanism, this subsection describes the execution process, focusing on the interaction between the high-level designer and the low-level executor during task solving.

DyFlow employs a layered control structure that separates high-level planning from low-level execution, ensuring adaptability in task-solving. This design aligns broad objectives with precise actions, using the state's context to inform decisions.

The **designer**, implemented as a policy $\pi_\theta$ with parameters $\theta$, manages strategic planning. At step $t$, it obtains a condensed state summary $f_{\text{summary}}(s_t)$ using GPT-4o-mini, then uses this summary to determine a subgoal and generate a stage subgraph $G_t$, selecting and configuring operator instances $(V_t)$ to achieve it. Formally, the designer generates:

$$G_t \sim \pi_\theta(\,\cdot \mid f_{\text{summary}}(s_t))\tag{1}$$

---

**Algorithm 1** DyFlow Framework for Complex Reasoning

---

**Require:** Task $p$, templates $\mathcal{O}$, designer $\pi_\theta$, executor $\pi_{\text{exec}}$, budget $T_{\max}$, summarizer $f_{\text{summary}}$
**Ensure:** Final answer $s_T$ and trajectory $\tau$

1:  $s_0 \leftarrow \{p\}$; $\tau \leftarrow [\,]$; $\mathcal{M} \leftarrow \{\}$         ▷ initial state, trace, and empty memory dictionary
2:  **for** $t = 0$ **to** $T_{\max} - 1$ **do**
3:     $z_t \leftarrow f_{\text{summary}}(s_t)$; $G_t \sim \pi_\theta(\cdot \,|\, z_t)$     ▷ summarize context and sample stage subgraph
4:     **for each** $o = (O_k, \phi, \psi) \in V_t$ (topological order) **do**
5:         Retrieve inputs $[\mathcal{M}[k] \mid k \in \psi]$       ▷ $\psi$ is a list of keys, $\mathcal{M}$ is a dictionary
6:         $r \leftarrow \pi_{\text{exec}}(\phi, [\mathcal{M}[k] \mid k \in \psi])$
7:         Generate a unique key $k_r$ for $r$       ▷ e.g., by operator id or execution order
8:         $\mathcal{M}[k_r] \leftarrow r$                  ▷ store output in memory dictionary
9:     **end for**
10:    $s_{t+1} \leftarrow \text{UpdateState}(s_t, G_t, \mathcal{M})$; $\tau.\text{append}((s_t, G_t))$     ▷ update state and record step
11:    **if** $C^t_{\text{end}}$ satisfied **or** TERMINATE **then**
12:       **break**                     ▷ stop if designer signals completion
13:    **end if**
14: **end for**
15: **return** $s_T$, $\tau$

---

where $G_t = (V_t, E_t, v^t_{\text{start}}, C^t_{\text{end}})$ and each operator instance $o \in V_t$ is defined as $o = (O_k, \phi, \psi)$ with $O_k \in \mathcal{O}$. The graph structure is designed to guide step-specific reasoning toward subgoal resolution.

Unlike prior frameworks such as AFlow, MaAS [12, 15], which implement control structures like branching or looping via explicit edge logic in code-based graphs, DyFlow adopts a designer-centric design. Instead of hardcoding control flow into the graph topology, we delegate all execution control to the designer. At each planning step, the designer observes the current state, including intermediate outputs, error signals, and planning history, and decides whether to proceed, revise, backtrack, or terminate. This allows DyFlow to simulate conditional and iterative behaviors (such as if-else statements or while loops) through repeated subgraph planning, without relying on manually defined edge logic. As a result, dynamic workflows emerge naturally from context-aware decision making, which simplifies graph construction and enhances flexibility across tasks.

The **workflow executor** translates this plan into action. Given $G_t$ and $s_t$, it carries out the operator instances in $V_t$ according to the dependencies in $E_t$, beginning with $v^t_{\text{start}} \in V_t$. For each operator $o = (O_k, \phi, \psi)$, the executor retrieves required inputs from the memory buffer $\mathcal{M}$ according to $\psi$, executes the operation via $\pi_{\text{exec}}$, and appends the result back to $\mathcal{M}$. The updated memory is then used to construct the next state $s_{t+1} = \text{Executor}(s_t, G_t)$. This mechanism ensures both consistent information flow and persistent memory accumulation across planning steps. The full procedure is summarized in Algorithm 1, which outlines the iterative interaction between the designer and executor.

### 3.3 Distilled Self-Play Learning

While the executor requires no additional training and can be directly instantiated using existing LLMs, the designer is trained separately to develop strong planning capabilities. We train the designer policy $\pi_\theta$ using a two-phase method that combines knowledge distillation with self-play preference optimization [22, 23, 24]. The goal is to enable the designer to construct context-sensitive subgoal plans that improve execution outcomes. While the executor remains fixed throughout, the designer is trained to dynamically generate structured reasoning workflows.

The learning process begins by generating trajectories $\tau = (s_0, G_0, s_1, G_1, \ldots, G_{T-1}, s_T)$ using the current or a past version of the policy $\pi_\theta$ across a diverse set of tasks. Each trajectory is executed by the workflow executor, and its outcome is compared to the correct solution to assess success. For successful trajectories, all stage-subgraph pairs $(f_{\text{summary}}(s_t), G_t)$ are labeled as "preferred," reflecting effective planning. For failed trajectories, the corresponding pairs are labeled as "discarded," capturing planning errors. All positive and negative examples are derived from these self-play trajectories, enabling the designer to learn from its own performance.

In the first phase, **knowledge distillation** initializes $\pi_\theta$ to generate reliable subgraphs. Using a dataset $D_{\text{SFT}} = \{(f_{\text{summary}}(s_t), G_t^{\text{expert}})\}$ of high-quality subgraphs from successful trajectories to optimize:

$$\mathcal{L}_{\text{SFT}}(\theta; D_{\text{SFT}}) = -\mathbb{E}_{(s, G^{\text{expert}}) \sim D_{\text{SFT}}} \left[ \log \pi_\theta(G^{\text{expert}} | f_{\text{summary}}(s)) \right] \tag{2}$$

This process distills knowledge from successful trajectories, producing the reference policy $\pi_{\text{ref}}$ for the next phase.

In the second phase, **self-play preference optimization** refines $\pi_\theta$ to favor subgraphs that lead to successful outcomes using KTO [23]. Using a dataset $D_{\text{pref}} = \{(f_{\text{summary}}(s_t), G_t, l_t)\}$, collected from self-generated trajectories produced by current or past policies, where $l_t \in \{\text{preferred}, \text{discarded}\}$ indicates whether the subgraph is positive or negative based on the trajectory's success, the optimization encourages effective planning:

$$\mathcal{L}_{\text{pref}}(\theta; D_{\text{pref}}, \pi_{\text{ref}}, \beta) = \mathbb{E}_{(s, G, l) \sim D_{\text{pref}}} \left[ L_{\text{pref}}^{\text{single}}(p_\theta, p_{\text{ref}}, l; \beta) \right]. \tag{3}$$

Here, $l \in \{\text{preferred}, \text{discarded}\}$ indicates whether the subgraph $G$ is effective or not, and $\beta$ is a hyperparameter controlling the strength of preference alignment. Here, $L_{\text{pref}}^{\text{single}}$ is computed based on whether each subgraph $G$ is labeled as preferred or discarded, with $\beta$ controlling the alignment strength between $\pi_\theta$ and $\pi_{\text{ref}}$. By leveraging feedback from self-generated trajectories, this off-policy self-play phase ensures the designer makes context-sensitive, adaptive decisions. This iterative self-improvement strategy equips DyFlow to deliver robust and versatile reasoning capabilities across a wide range of applications.

**Preference Optimization Strategy.** We adopt trajectory-level supervision for preference optimization, using full execution success or failure as the training signal. Assigning fine-grained rewards to individual subgraphs is unreliable in complex reasoning tasks, where even well-formed plans may fail due to executor variability. For similar reasons, we avoid DPO-style pairwise ranking [24], as it is often infeasible to construct clear positive–negative plan pairs: execution outcomes depend on downstream decisions, and the better plan is not always evident from intermediate states. Online reinforcement learning poses additional challenges, reward signals are typically sparse and delayed, and using LLM-based judges introduces high variance due to inconsistent evaluations across steps.[25] In contrast, our offline self-play setup with KTO [23] provides a more stable learning process, enabling effective designer refinement from diverse execution trajectories.

## 4    Experiments

We conduct comprehensive experiments to evaluate DyFlow across five reasoning domains. Our goals are to assess its performance advantage over existing agent planning frameworks, its generalization capabilities to unseen datasets and executor models, the contribution of each component through ablation studies, and qualitative behaviors through case analyses.

### 4.1    Experimental Setup

**Datasets.** We consider 5 diverse reasoning domains, each represented by a benchmark dataset: (1) **Logical Reasoning** using the LiveBench dataset [26], (2) **Math Reasoning** using the MATH benchmark [27], (3) **Medical Reasoning** with PubMedQA [4], (4) **Code Reasoning** via HumanEval [28], and (5) **Social Reasoning** using the SocialMaze [29] benchmark. These datasets differ in their input structures, intermediate reasoning steps, and solution formats, and together cover a broad spectrum of reasoning behaviors.

**Implement Details.** DyFlow is trained only on MATH, PubMedQA, and LiveBench. HumanEval and SocialMaze are held out to assess zero-shot generalization to unseen reasoning domains. To avoid contamination and ensure valid knowledge separation between the designer and executor, we use Phi-4 [30] as the executor model, and train a designer we refer to as **DyPlanner**, which is initialized from Phi-4 and optimized using the DyFlow training framework. The initial distillation during pretraining is conducted using GPT-4.1 [31], based on trajectories generated only from our training set. At inference time, both the designer and the executor are run with temperature 0.01 to ensure stable and deterministic behaviors.

Table 1: Performance comparison across five reasoning domains. All methods use `phi-4` as the executor. **Bold** marks the best score in each column. The figures in ↑ / ↓ indicate the absolute change with respect to the *Vanilla* baseline.

| Method | SocialMaze | PubMedQA | MATH | LiveBench | HumanEval | Avg. |
|---|---|---|---|---|---|---|
| Vanilla | 6.49 | 67.33 | 66.80 | 40.00 | 86.59 | 53.44 |
| CoT [32] | 6.11↓0.38 | 67.73↑0.40 | 71.20↑4.40 | 39.33↓0.67 | 87.80↑1.21 | 54.43↑0.99 |
| SC [33] | 10.31↑3.82 | 68.53↑1.20 | 71.60↑4.80 | 42.00↑2.00 | 87.20↑0.61 | 55.93↑2.49 |
| LLM-Debate [8] | 12.59↑6.10 | 68.53↑1.20 | 72.40↑5.60 | 41.33↑1.33 | 84.15↓2.44 | 55.80↑2.36 |
| Self-Refine [34] | 10.69↑4.20 | 69.32↑1.99 | 70.40↑3.60 | 34.67↓5.33 | 81.71↓4.88 | 53.36↓0.08 |
| ADAS [13] | 6.11↓0.38 | 67.33↑0.00 | 66.80↑0.00 | 39.33↓0.67 | 85.37↓1.22 | 52.99↓0.45 |
| AFlow [12] | 11.45↑4.96 | 69.72↑2.39 | 74.00↑7.20 | 43.33↑3.33 | 89.02↑2.43 | 57.50↑4.06 |
| MaAS [15] | 13.36↑6.87 | 69.32↑1.99 | 73.60↑6.80 | 44.00↑4.00 | 88.41↑1.82 | 57.74↑4.30 |
| **DyFlow (Ours)** | **17.18**↑10.69 | **72.91**↑5.58 | **76.40**↑9.60 | **48.67**↑8.67 | **92.07**↑5.48 | **61.45**↑8.01 |

**Baselines.** We compare DyFlow with a comprehensive set of baselines across two categories: (1) *Prompting-based methods*, including **Vanilla** prompting, **CoT** [32], **Self-Consistency (SC)** [33], **LLM-Debate** [8], and **Self-Refine** [34]. (2) *Automated Agent frameworks*, including **ADAS** [13], **AFlow** [12], and **MaAS** [15], which introduce structured reasoning workflows with varying degrees of adaptivity. All methods use the same executor (Phi-4) and operator set for fair comparison, and are evaluated under consistent metrics: accuracy for most tasks and pass@1 for code reasoning.

### 4.1.1 Performance Comparison

**Overall Performance** Table 1 reports the performance of DyFlow and all baselines across five reasoning domains. DyFlow consistently outperforms prior methods, achieving the highest average accuracy (61.45) with improvements observed across logic, math, medical, code, and social reasoning tasks. Notably, while trained only on MATH, PubMedQA, and LiveBench, DyFlow generalizes robustly to the held-out HumanEval and SocialMaze benchmarks. These results highlight the transferability of DyFlow's adaptive planning strategy across structurally diverse domains. In particular, DyFlow achieves 17.18 on SocialMaze, a challenging multi-turn benchmark that requires high-level reasoning, substantially outperforming all baselines in this zero-shot setting. This strong performance demonstrates that DyFlow's feedback-driven, adaptive planning enables effective subgoal revision and robust reasoning, even on previously unseen, highly complex tasks.

**Planning Upper Bound and Stability** In addition to Pass@1, we further evaluate the performance of DyFlow under the top-k setting. As shown in Figure 3, DyFlow consistently outperforms CoT across all $k$ values from 1 to 5, with increasingly wider margins at higher $k$. This indicates that DyFlow not only generates more accurate first predictions, but also demonstrates greater reasoning stability across multiple completions, with a higher potential to produce strong solutions.

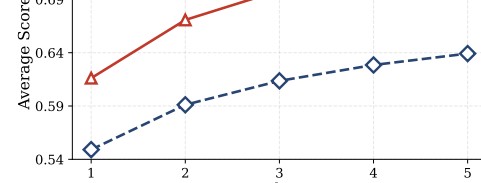

Figure 3: Average pass@k comparisons between DyFlow and CoT across 5 benchmarks.

Notably, DyFlow reaches a near-perfect Pass@5 of 0.9817 on HumanEval, highlighting the framework's strong reasoning upper bound on code tasks. Moreover, the consistent improvements in medical, logical, and math domains suggest that DyFlow enhances both accuracy and diversity of reasoning paths. Per-task breakdowns of Pass@k curves are included in Figure 7, which reveal similar trends across all domains.

Table 2: Performance of DyFlow with different designer models across five reasoning domains. All designers are trained using the same pipeline. Detailed cost of different designers is in Figure 9.

| Designer Model | SocialMaze | PubMedQA | MATH | LiveBench | HumanEval | Average |
|---|---|---|---|---|---|---|
| MaAS [15] | 13.36 | 69.32 | 73.60 | 44.00 | 88.41 | 58.40 |
| Claude-3.7-Sonnet | 16.41 | 71.71 | 75.60 | **50.00** | **92.07** | 61.16 |
| GPT-4.1 | 16.79 | 72.11 | 73.60 | 47.03 | 89.63 | 59.83 |
| **DyPlanner** | **17.18** | **72.91** | **76.40** | 48.67 | **92.07** | **61.45** |

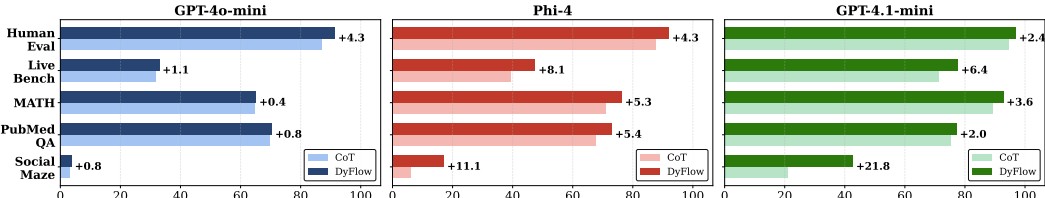

Figure 4: Cross-executor performance comparison between CoT and DyFlow across five reasoning tasks. DyFlow improves performance across all model scales, with larger gains for stronger models on more challenging tasks. Detailed results are provided in Appendix D.5, Table 10.

## 4.2 Generalization Analysis

This section evaluates DyFlow's generalization capabilities across three dimensions: designer models, execution models, and datasets. Each dimension demonstrates DyFlow's ability to maintain robust performance under varying conditions, leveraging its adaptive planning mechanisms.

**Cross-Designer Generalization**    To assess the robustness of our training framework across different designer architectures, we apply the same DyFlow learning pipeline to three backbone models: Claude-3.7-Sonnet, GPT-4.1, and Phi-4. We include MaAS as the strongest existing baseline.

As shown in Table 2, DyFlow consistently outperforms MaAS across all designer backbones. More specifically, our DyPlanner is initialized from the 14B open-weight Phi-4 model. Despite its relatively small scale, DyPlanner achieves performance comparable to larger proprietary designers such as GPT-4.1 and Claude-3.7-Sonnet across most domains. This demonstrates that DyFlow's strong planning capability does not stem from model size alone, but also from our dedicated two-phase optimization strategy. By combining supervised subgraph distillation with offline preference-based refinement, we equip a compact model with the ability to perform high-quality structured planning across diverse tasks. As shown in Table 9, DyPlanner offers a more cost-efficient and scalable alternative without sacrificing performance. These results suggest that dynamic, feedback-aware designers can emerge from lightweight open models when trained with appropriate trajectory-level signals, making DyFlow well-suited for real-world deployment in resource-constrained settings.

**Cross-Executor Generalization**    To further evaluate DyFlow's generalization capability across executor models, we compare its performance with CoT prompting using three different executors: GPT-4o-mini, Phi-4, and GPT-4.1-mini. All DyFlow variants in this setting use the same DyPlanner. As shown in Figure 4, DyFlow consistently improves performance across all executors and reasoning domains, demonstrating its compatibility with a wide range of language models without requiring executor-specific adaptation.

Table 3: Cross-task generalization. Each test domain is excluded during training.

| Test | Train | CoT | DyFlow |
|---|---|---|---|
| SR | MR, QR, LR | 6.11 | **17.18** |
| CR | MR, QR, LR | 87.80 | **92.07** |
| MR | CR, LR, SR | 71.08 | **75.20** |
| QR | CR, LR, SR | 67.73 | **72.11** |

The gains are especially notable when DyFlow is paired with stronger executors and applied to more complex tasks. For instance, Phi-4 with DyFlow achieves performance close to GPT-4.1-mini with CoT, while incurring only half the cost. When applied to GPT-4.1-mini, DyFlow brings further improvements, particularly on SocialMaze and LiveBench. These results indicate that structured

planning from DyPlanner enhances reasoning quality and stability across a variety of executor backbones.

**Cross-Task Generalization**    To evaluate cross-task generalization, we construct a series of held-out settings where the designer is trained on a subset of three reasoning domains and tested on the remaining two. For each setting, we select three domains from SocialMaze, PubMedQA, MATH, LiveBench, and HumanEval for training, and evaluate performance on the two excluded domains. For example, when training on MATH, PubMedQA, and LiveBench, we test on SocialMaze and HumanEval. Similarly, other combinations ensure that each domain is excluded from training in at least one setting. As shown in Table 3, DyFlow achieves strong performance across the held-out domains, demonstrating that its designer can generalize planning strategies to structurally diverse, unseen reasoning tasks without task-specific supervision.

Table 4: Ablation study on DyFlow using DyPlanner as the designer and Phi-4 as the executor. The full system outperforms all ablated variants across five reasoning tasks.

| Variant | SocialMaze | PubMedQA | MATH | LiveBench | HumanEval | Avg. |
|---|---|---|---|---|---|---|
| w/o KTO | 13.36 | 68.53 | 72.40 | 43.33 | 89.63 | 57.45 |
| w/o SFT | 15.65 | 69.32 | 73.20 | 45.33 | 91.46 | 59.00 |
| w/o Dynamic Operator | 12.21 | 68.92 | 72.80 | 40.67 | 90.24 | 56.97 |
| w/o Dynamic Planning | 11.45 | 68.53 | 72.80 | 40.00 | 89.63 | 56.48 |
| **DyFlow (Full)** | **17.18** | **72.91** | **76.40** | **48.67** | **92.07** | **61.45** |

## 4.3   Ablation Study

To assess the contribution of DyFlow's core components, we conduct ablation experiments targeting both the designer training process and the execution-time mechanisms, as summarized in Table 4. On the training side, we remove either the knowledge distillation (SFT) or the offline preference optimization (KTO). Both lead to consistent drops in performance, indicating that these two phases are complementary: SFT provides structural guidance for initializing the designer, while KTO enables refinement based on trajectory-level feedback. On the execution side, disabling dynamic operator selection forces the executor to follow a fixed-stage template, reducing its ability to adjust reasoning based on intermediate results.

The largest degradation is observed when dynamic planning is removed. In this setting, we prompt DyPlanner to complete the entire problem in a single reasoning stage without receiving intermediate feedback. This design prevents the designer from iteratively refining subgoals in response to execution signals, undermining DyFlow's central mechanism of feedback-driven replanning. Together, these results demonstrate that DyFlow's gains emerge from the synergy between structured designer training and dynamic subgoal adaptation during execution.

## 4.4   Case Study

We conduct a qualitative comparison between DyFlow and a standard reasoning baseline without error correction. The baseline generates a fixed reasoning path based on initial inputs and executes it without revisiting intermediate outputs, making it susceptible to early mistakes and incomplete solutions. As illustrated in Figure 5, DyFlow instead treats planning as an ongoing process. Guided by the designer, it dynamically assigns operators based on intermediate feedback and task state. When execution

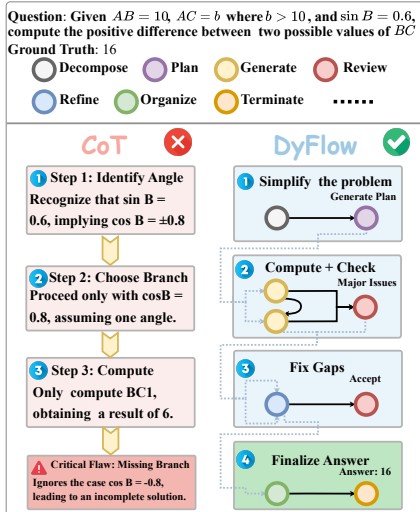

Figure 5: Case Study between CoT and DyFlow on MATH dataset.

produces partial or inconsistent outputs, DyFlow adapts its subgoal plan and invokes refinement operators to identify and correct reasoning errors during runtime. This includes dynamically selecting

the appropriate data inputs, adjusting instructions, and refining intermediate results using targeted operator calls.

## 5 Conclusion

In this paper, we presented DyFlow, a framework for dynamic workflow construction in LLM-based reasoning systems. By modeling reasoning as subgraph planning with modular operator execution, DyFlow enables flexible adaptation to intermediate feedback and diverse task requirements. Our evaluations across five reasoning domains demonstrate that DyFlow improves task success rates and generalizes across diverse reasoning tasks. Future work may explore integrating richer error detection signals and expanding the operator set to support more complex interaction protocols.

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

# A   Notations

We summarize the key notations used throughout the DyFlow framework in Table 5. These notations cover the planning state space, operator structure, execution dynamics, and training objectives for optimizing the designer policy.

Table 5: Key notations used in the DyFlow framework.

| Symbol | Definition |
|---|---|
| $p$ | A reasoning task sampled from a task distribution. |
| $s_t$ | Full system state at discrete step $t$, including task inputs and execution history. |
| $\mathcal{O}$ | A fixed set of operator templates specifying reusable functional behaviors. |
| $G_t$ | Stage subgraph constructed at step $t$: $G_t = (V_t, E_t, v_{\text{start}}^t, C_{\text{end}}^t)$. |
| $o$ | An operator instance, formally represented as $o = (O_k, \phi, \psi)$ with $O_k \in \mathcal{O}$. |
| $\phi$ | Fine-grained instruction string instantiated from template $O_k$ based on context. |
| $\psi$ | Input reference list indexing outputs from memory buffer $M$. |
| $\mathcal{M}$ | Global memory storing intermediate outputs generated by executed operators. |
| $\pi_\theta$ | Learnable planner (designer) policy that generates $G_t$ from current state summary. |
| $\theta$ | Parameters of the planner policy $\pi_\theta$ trained via SFT and KTO. |
| $f_{\text{summary}}(s_t)$ | Function that summarizes full state $s_t$ into a condensed representation. |
| $z_t$ | Summarized context at step $t$: $z_t = f_{\text{summary}}(s_t)$. |
| $\pi_{\text{exec}}$ | Fixed executor policy used to run operator instances during workflow execution. |
| $\tau$ | Full execution trajectory: $\tau = (s_0, G_0, s_1, G_1, \ldots, s_T)$. |
| $L_{\text{SFT}}$ | Supervised fine-tuning loss used to initialize the planner $\pi_\theta$. |
| $D_{\text{SFT}}$ | Dataset of successful planning trajectories used for supervised training. |
| $\pi_{\text{ref}}$ | Reference planner derived from SFT, used during preference optimization. |
| $L_{\text{pref}}$ | Preference optimization loss (e.g., KTO), aligning $\pi_\theta$ with effective plans. |
| $D_{\text{pref}}$ | Labeled trajectory set used for preference-based policy refinement. |

# B   Theoretical Analysis

**DyFlow is never worse than static method**   We consider a distribution $\mathcal{D}$ over reasoning tasks $p$. Each task is solved by executing a sequence of stage subgraphs $\{G_t\}_{t=0}^{T-1}$, yielding return $R(p; \pi)$. Define Equation 4 as objective function.

$$J(\pi) = \mathbb{E}_{p \sim \mathcal{D}}\big[R(p; \pi)\big] \tag{4}$$

Each reasoning task $p$ is modeled as a finite-horizon decision process $\langle \mathcal{S}, \mathcal{O}, P, r, T \rangle$ with bounded rewards, finite state space $\mathcal{S}$, and operator set $\mathcal{O}$. A static policy $\pi_{\text{stat}}$ applies the same subgraph $G_{\text{fix}}$ throughout regardless of context (i.e. $\pi_{\text{stat}}(s_t) \equiv G_{\text{fix}}$), while the DyFlow policy $\pi_{\text{DyFlow}}$ chooses $G_t$ as described in Equation 5, based on the full state $s_t$, including intermediate outputs and error signals.

$$G_t = \pi_{\text{DyFlow}}\big(f_{\text{summary}}(s_t)\big) \tag{5}$$

**Lemma 1** (Static Policies as a Special Case). $\Pi_{\text{stat}} \subseteq \Pi_{\text{DyFlow}}$, *as any static $\pi_{\text{stat}}$ can be implemented by DyFlow by ignoring $s_t$ and always returning $G_{\text{fix}}$.*

**Theorem 1** (DyFlow Is Never Worse Than Static).

$$\max_{\pi \in \Pi_{\text{DyFlow}}} J(\pi) \geq \max_{\pi \in \Pi_{\text{stat}}} J(\pi). \tag{6}$$

*Moreover, if there exists a task $p_0$ for which feedback-driven replanning strictly improves return, then*

$$\max_{\pi \in \Pi_{\text{DyFlow}}} J(\pi) > \max_{\pi \in \Pi_{\text{stat}}} J(\pi). \tag{7}$$

**Proof 1.** *Since $\Pi_{\text{stat}} \subseteq \Pi_{\text{DyFlow}}$ by Lemma 1, optimizing over the larger set $\Pi_{\text{DyFlow}}$ can only increase (or match) the best static return like Equation 8.*

$$\max_{\pi \in \Pi_{\text{DyFlow}}} J(\pi) \geq \max_{\pi \in \Pi_{\text{stat}}} J(\pi) \tag{8}$$

*In stochastic or error-prone environments, a closed-loop DyFlow policy can correct deviations in real time—e.g. via replanning—yielding strictly higher return on some task $p_0$ [35]. Hence the inequality becomes strict.*

**Convergence proof of DyFlow**   We model each reasoning task $p$ as a finite-horizon MDP of length $T$. For $t = 0, 1, \ldots, T$, let $V_t^*(s)$, $V_t^{\text{Dy}}(s)$ denote the optimal and DyFlow value functions when there are $t$ steps remaining from state $s$. At step $t$, DyFlow observes $s_t$, computes the summary $z_t = f_{\text{summary}}(s_t)$, and samples a subgraph $G_t \sim \pi_{\text{DyFlow}}(\cdot \mid z_t)$. Executing $G_t$ yields reward $r(s_t, G_t)$ and transitions to $s_{t+1}$.

One-step Bellman operator can be operated for such MDP process with DyFlow one-step lookup feature, denoted as Equation 9

$$(\mathcal{T}_t V)(s; G) \; = \; r(s, G) \; + \; \mathbb{E}_{s' \sim P(\cdot | s, G)}\big[V(s')\big] \tag{9}$$

and quantify DyFlow's per-step suboptimality by the Bellman residual

$$\delta_t \; = \; \max_s \Big| \underbrace{\max_G \mathcal{T}_t V_{t-1}^{\text{Dy}}(s; G)}_{\text{optimal backup at step } t} \; - \; \underbrace{\mathcal{T}_t V_{t-1}^{\text{Dy}}(s; G_t)}_{\text{DyFlow's backup at step } t} \Big|. \tag{10}$$

Let $\varepsilon_t \geq \delta_t$ be a uniform upper bound on this residual.

**Lemma 2** (Error propagation). *For any $t = 0, 1, \ldots, T$ and state $s$, the gap between the optimal and DyFlow value functions with $t$ steps remaining satisfies Equation 11.*

$$V_t^*(s) \; - \; V_t^{\text{Dy}}(s) \; \leq \; \sum_{k=1}^{t} \varepsilon_k \tag{11}$$

*Proof.* **Case 1.** With zero steps remaining, both $V_0^*(s)$ and $V_0^{\text{Dy}}(s)$ equal the terminal reward (here normalized to 0), so the inequality holds.

**Case 2.** Assume the claim for $t - 1$. Then for horizon $t$,

$$V_t^*(s) - V_t^{\text{Dy}}(s) = \max_G \mathcal{T}_t V_{t-1}^*(s; G) \; - \; \mathcal{T}_t V_{t-1}^{\text{Dy}}(s; G_t).$$

Split into two parts:

$$\underbrace{\max_G \mathcal{T}_t V_{t-1}^{\text{Dy}}(s; G) - \mathcal{T}_t V_{t-1}^{\text{Dy}}(s; G_t)}_{\leq \, \delta_t \leq \varepsilon_t} \; + \; \underbrace{\max_G \mathcal{T}_t V_{t-1}^*(s; G) - \max_G \mathcal{T}_t V_{t-1}^{\text{Dy}}(s; G)}_{\leq \, \sup_{s'} \left[ V_{t-1}^*(s') - V_{t-1}^{\text{Dy}}(s') \right]}.$$

By the inductive hypothesis, $\sup_{s'} \left[ V_{t-1}^*(s') - V_{t-1}^{\text{Dy}}(s') \right] \leq \sum_{k=1}^{t-1} \varepsilon_k$. Hence

$$V_t^*(s) - V_t^{\text{Dy}}(s) \; \leq \; \varepsilon_t \; + \; \sum_{k=1}^{t-1} \varepsilon_k \; = \; \sum_{k=1}^{t} \varepsilon_k,$$

completing the induction. $\square$

**Theorem 2** (DyFlow performance bound). *Starting from initial state $s_0$ with full horizon $T$ along with Lemma 2,*

$$V_T^*(s_0) \; - \; V_T^{\text{Dy}}(s_0) \; \leq \; \sum_{k=1}^{T} \varepsilon_k \; \leq \; T \max_{1 \leq k \leq T} \varepsilon_k.$$

## C   Operators

DyFlow employs modular and reusable operator templates to support flexible reasoning workflows. Each operator defines a specific functional role and is instantiated dynamically by the planner during execution. We summarize all templates used in DyFlow and analyze their usage patterns across reasoning domains.

## C.1 Operator Templates

Table 6 summarizes all operator templates used by DyFlow. Each template corresponds to a modular and reusable functional unit. During execution, the designer instantiates these templates with fine-grained instructions ($\phi$) and dynamic inputs ($\psi$) based on the current context.

Table 6: DyFlow operator templates and their functional roles.

| Template Name | Description |
|---|---|
| GENERATE_PLAN | Propose a high-level plan for solving the current subgoal. |
| DECOMPOSE_PROBLEM | Break a complex goal into subgoals. |
| GENERATE_ANSWER | Produce an answer candidate for the current task. |
| REVIEW_SOLUTION | Evaluate correctness or completeness of a prior answer. |
| REFINE_ANSWER | Modify or improve a previously generated answer. |
| GENERATE_CODE | Write code to solve the current subgoal. |
| REFINE_CODE | Improve or debug previously generated code. |
| ORGANIZE_SOLUTION | Summarize or structure the final answer for output. |
| ENSEMBLE | Aggregate multiple reasoning paths using voting. |
| DEFAULT | General-purpose fallback operator. |
| TERMINATE | End the workflow once the solution is complete. |

## C.2 Operator Usage Analysis

Figure 6 summarizes the frequency of operator usage across different reasoning domains. We observe that operators like REVIEW_SOLUTION, TERMINATE, and ORGANIZE_SOLUTION appear frequently across all tasks, reflecting their general-purpose utility. In contrast, operators such as DECOMPOSE_PROBLEM, REFINE_ANSWER, and GENERATE_PLAN are more selectively used in complex domains like LiveBench and SocialMaze, where dynamic adaptation and structure restructuring are more critical.

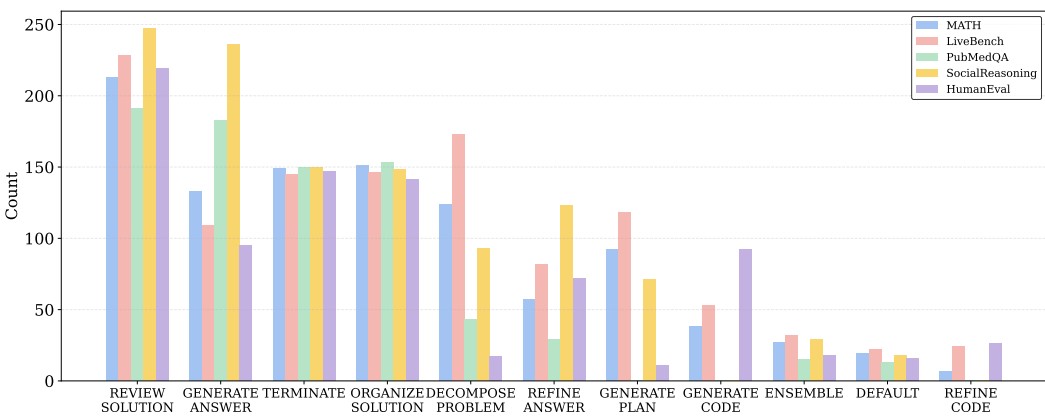

Figure 6: Operator usage frequency across reasoning domains. Task-specific patterns reflect how DyFlow dynamically adjusts workflow structure based on domain complexity and intermediate feedback.

# D Experiment Details

## D.1 Dataset Statistics

Table 7 summarizes the dataset statistics used in our experiments. We adopt a default TRAIN:TEST split ratio of approximately 1:3 across all datasets to balance supervision and evaluation coverage. For the MATH benchmark, we follow the setting in MaAS [36], selecting problems at difficulty level 5 across four representative categories: Combinatorics & Probability, Number Theory, Pre-algebra, and Pre-calculus. For HumanEval, when it is not included in the training set, we evaluate on the full 164 problems to ensure consistency with prior work.

Table 7: Train/test statistics for each dataset.

| Domain | Dataset | #Train | #Test | Metric |
|---|---|---|---|---|
| Social Reasoning | SocialMaze | 87 | 262 | Accuracy |
| Medical Reasoning | PubMedQA | 84 | 251 | Accuracy |
| Math Reasoning | MATH | 84 | 250 | Accuracy |
| Logic Reasoning | LiveBench | 50 | 150 | Accuracy |
| Code Reasoning | HumanEval | 55 | 109 | pass@1 |

## D.2 Designer Training

We use Phi-4 [30] as the designer policy $\pi_\theta$, trained on 2 Nvidia A6000 GPUs with LoRA-based parameter-efficient tuning [37]. Training proceeds in two stages. First, supervised fine-tuning is performed on 1.5k design results from MATH, PubMedQA, and Livebench, using a cutoff length of 2048, batch size 1 with gradient accumulation steps of 4, learning rate $5 \times 10^{-6}$, cosine learning rate scheduler with warmup ratio 0.1, bf16 precision, and 3 training epochs. The validation split is 10%.

Second, we apply KTO [23] for preference-based refinement using 2k design results with a 1:1 positive-to-negative ratio, labeled based on task success. This stage uses a cutoff length of 4096, batch size 1 with gradient accumulation steps of 8, learning rate $2 \times 10^{-4}$, KL penalty $\beta = 0.1$, bf16 precision, cosine scheduler, and 3 epochs. Validation is performed every 500 steps. Each trajectory includes a task description, the planned subgraphs, intermediate operator outputs, and the final answer.

## D.3 Inference and Evaluation

At inference time, both the designer and executor are run with temperature 0.01 to ensure deterministic outputs. All methods use Phi-4 to ensure fair comparison.

## D.4 Cost Analysis

We report the cost of DyFlow in both training and inference, and compare it with baseline systems using proprietary or open-weight models. As shown in Table 9, DyPlanner achieves the lowest inference cost across all domains while maintaining comparable or superior performance to larger proprietary designers such as GPT-4.1 and Claude-3.7. Despite being based on the compact Phi-4 model, DyPlanner enables DyFlow to match or outperform these stronger backbones in structured reasoning tasks. This confirms the effectiveness of our training pipeline, which distills high-quality planning behavior into a lightweight designer with minimal runtime cost.

We further analyze the full pipeline cost in Table 8, including both designer and executor token usage during training and inference on the MATH benchmark. Although DyFlow incurs additional training cost due to initial distillation from GPT-4.1, it achieves the highest performance among all methods. During inference, DyFlow's total token cost is moderately higher than AFlow and MaAS (approximately 1.4x to 3x), owing to its two-stage designer–executor structure. However, this cost remains substantially lower than prompt-based methods such as LLM-Debate and Self-Refine, which lack structural planning. These results suggest that DyFlow offers a favorable trade-off: higher

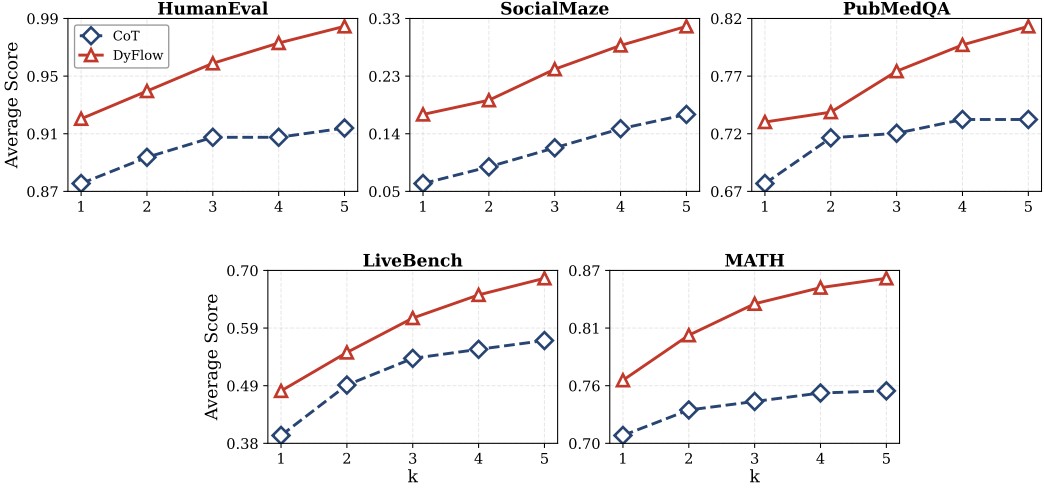

Figure 7: Per-benchmark Pass@k comparisons between DyFlow and CoT across five reasoning domains. DyFlow consistently achieves better accuracy under all $k$ values in each task.

Table 8: Cost and performance comparison on the MATH dataset. DyFlow is trained only on MATH, PubMedQA, and LiveBench; Dyflow training cost is computed using only the MATH portion. **Bold** marks the best (lowest for cost/tokens, highest for accuracy).

| Method | Training | | | Inference | | | Acc. (%) |
|--------|---------|-------|---------|---------|-------|---------|---------|
| | **Prompt** | **Comp.** | **Cost ($)** | **Prompt** | **Comp.** | **Cost ($)** | |
| LLM-Debate | – | – | – | 1,449,574 | 4,859,777 | 0.78 | 72.40 |
| Self-Refine | – | – | – | 2,498,569 | 1,768,407 | 0.42 | 70.40 |
| AFlow | 13,773,792 | 13,240,624 | 2.82 | 1,208,685 | 895,017 | 0.21 | 74.00 |
| MaAS | 1,445,257 | 1,011,317 | **0.24** | **528,145** | **419,009** | **0.10** | 73.60 |
| **DyFlow (D.)** | **788,910** | **297,669** | 3.96 | 773,625 | 225,737 | 0.09 | **76.40** |
| **DyFlow (E.)** | **617,560** | **452,044** | 0.11 | 1,288,918 | 881,452 | 0.21 | |

reasoning accuracy and generalization, with only modest overhead compared to other workflow-based systems.

### D.5 Cross-Executor Performance Details

We provide detailed performance comparisons of CoT and DyFlow across three executor models and five reasoning tasks in Table 10. DyFlow consistently improves performance over CoT under all configurations, demonstrating its robustness to executor variation. The gains are particularly notable for more lightweight models such as Phi-4 and GPT-4o-mini, where CoT struggles with reasoning consistency. In contrast, DyFlow's explicit planning compensates for executor limitations, leading to substantial improvements in domains such as SocialMaze and MATH.

## E   Limitations

Although DyFlow demonstrates strong generalization and reasoning stability across a variety of tasks, its current design has a notable limitation: the lack of integration with external tools and APIs. Specifically, DyFlow's operator set is primarily built for symbolic reasoning (e.g., mathematical derivation, code execution) and textual reasoning (e.g., question answering), and does not yet support operations such as search engine access, database queries, or environment interaction. This limitation can lead to performance bottlenecks in tasks that require complex tool-assisted reasoning.

Table 9: Input tokens, output tokens, and computational costs of DyFlow with different designer models across five reasoning domains.

| Benchmark | Designer | Input Tokens | Output Tokens | Cost (USD) |
|---|---|---|---|---|
| HumanEval | GPT-4.1 | 517575 | 211600 | 2.73 |
| | Claude-3.7-Sonnet | 511347 | 205813 | 4.62 |
| | **DyPlanner** | 505603 | 199355 | **0.06** |
| LiveBench | GPT-4.1 | 815952 | 275565 | 3.84 |
| | Claude-3.7-Sonnet | 803211 | 271937 | 6.49 |
| | **DyPlanner** | 791525 | 300402 | **0.10** |
| MATH | GPT-4.1 | 788910 | 297669 | 3.96 |
| | Claude-3.7-Sonnet | 781589 | 253471 | 6.15 |
| | **DyPlanner** | 773625 | 225737 | **0.09** |
| PubMedQA | GPT-4.1 | 911364 | 302684 | 4.24 |
| | Claude-3.7-Sonnet | 887653 | 289117 | 7.00 |
| | **DyPlanner** | 588259 | 213749 | **0.07** |
| SocialMaze | GPT-4.1 | 1499141 | 436371 | 6.49 |
| | Claude-3.7-Sonnet | 1387923 | 412689 | 10.35 |
| | **DyPlanner** | 905088 | 300554 | **0.11** |
| **Total** | GPT-4.1 | 4532942 | 1523889 | 21.26 |
| | Claude-3.7-Sonnet | 4371723 | 1433027 | 34.61 |
| | **DyPlanner** | 3564100 | 1239797 | **0.42** |

Table 10: Detailed scores of CoT and DyFlow across three executor models and five reasoning tasks. Best results in each column are highlighted in **bold**.

| Model | SocialMaze | PubMedQA | MATH | LiveBench | HumanEval |
|---|---|---|---|---|---|
| GPT-4o-mini (CoT) | 3.05 | 69.72 | 64.80 | 32.67 | 87.20 |
| GPT-4o-mini (DyFlow) | **3.82** | **70.52** | **65.20** | **32.67** | **91.46** |
| Phi-4 (CoT) | 6.11 | 67.73 | 71.08 | 39.33 | 87.80 |
| Phi-4 (DyFlow) | **17.18** | **72.91** | **76.40** | **48.67** | **92.07** |
| GPT-4.1-mini (CoT) | 20.99 | 75.30 | 89.20 | 66.00 | 94.51 |
| GPT-4.1-mini (DyFlow) | **42.75** | **77.29** | **92.80** | **71.33** | **96.95** |

It is worth noting, however, that in most of the tasks we evaluate, performance bottlenecks arise mainly from subgoal planning or logical reasoning errors rather than from missing external knowledge or tool usage, which is consistent with observations in prior research [38]. As such, within the scope of our current evaluation, DyFlow's structured planning and feedback mechanisms already lead to significant performance gains, and the lack of tool integration has only limited impact.

Nevertheless, as tasks continue to expand into more complex domains—such as web-based question answering, embodied interaction, and multimodal reasoning—the demand for external tool use is expected to increase. Future work may explore abstracting tool operations as special operator types and extending the executor interface, thereby enhancing DyFlow's generality and applicability in real-world, complex environments.

# F   Case Study

## Case Study (MATH)

### Problem Description

**Task:** Given $AB = 10$, $AC = b$ where $b > 10$, and $\sin B = \frac{3}{5}$, compute the positive difference between the two possible values of $BC$.

### CoT Answer by Phi-4

**Step 1:** Recognize that $\sin B = \frac{3}{5} \Rightarrow \cos B = \pm\frac{4}{5}$.
**Step 2:** Proceed with $\cos B = \frac{4}{5}$ (acute angle).
**Step 3:** Apply Law of Cosines to compute $BC_1$ and output result.
**Critical Flaw:** The solution ignores the second case $(\cos B = -\frac{4}{5})$, which leads to an incomplete answer.
**Final Answer:** 6                                          **Incorrect**

### DyFlow Answer by Phi-4

**Stage 1: Planning and Decomposition**
Operator: `DECOMPOSE_PROBLEM`
Output: Identifies $\cos B = \pm\frac{4}{5}$ as two geometric branches

**Stage 2: Partial Answer Generation + Initial Review**
Operator: `GENERATE_ANSWER` — Computes $BC_1$ for $\cos B = \frac{4}{5}$
Operator: `GENERATE_ANSWER` — Computes $BC_2$ for $\cos B = -\frac{4}{5}$
Operator: `REVIEW_SOLUTION` — Aggregates both results, checks completeness

**Stage 3: Refinement and Re-evaluation**
Operator: `REFINE_ANSWER` — Corrects inconsistency or adds missing branches
Operator: `REVIEW_SOLUTION` — Validates the refined answer

**Stage 4: Finalization**
Operator: `ORGANIZE_SOLUTION`, `TERMINATE`

**Final Answer: 16**                                          **Correct**

**Case Study (Livebench)**

**Problem Description**

**Task:** How many ordered pairs of positive integers $(m, n)$ satisfy $\gcd(m, n) = 2$ and $\text{lcm}[m, n] = 108$?

**Ground Truth: 4**

**CoT Answer by Phi-4**

**Steps:**

- Use identity $mn = \gcd(m, n) \cdot \text{lcm}[m, n] \Rightarrow mn = 216$.
- Let $m = 2a$, $n = 2b$ with $\gcd(a, b) = 1 \Rightarrow ab = 54$.
- Only considers $(a, b) = (1, 54), (54, 1)$, which gives $(m, n) = (2, 108), (108, 2)$.

**Flaw:** Misses other coprime factorizations of 54, such as $(2, 27)$ and $(27, 2)$, leading to incomplete counting.

**Final Answer: 2**                                                                 **Incorrect**

**DyFlow Answer by Phi-4**

**Stage 1: Decomposition and Initial Attempt**
DECOMPOSE_PROBLEM, GENERATE_ANSWER, REVIEW_SOLUTION
Uses $m = 2a$, $n = 2b$ with $\gcd(a, b) = 1$ and $ab = 54$, and systematically identifies all valid $(a, b)$ pairs by distributing the prime powers $2^1$ and $3^3$ disjointly between $a$ and $b$.
**Valid $(a, b)$ pairs:** $(1, 54), (2, 27), (27, 2), (54, 1)$
**Corresponding $(m, n)$:** $(2, 108), (4, 54), (54, 4), (108, 2)$

**Stage 2: Finalization**
ORGANIZE_SOLUTION, TERMINATE
Verifies all pairs meet both $\gcd = 2$ and $\text{lcm} = 108$.

**Final Answer: 4**                                                                     **Correct**

## Case Study (Logic Puzzle)

### Problem Description

**Task:** In this question, assume each person either always tells the truth or always lies. [Details of people, locations, and statements are given]. Does the person at the farm tell the truth? Does the person at the restaurant tell the truth? Does the person at the observatory tell the truth? Output answer as a list of three words: yes or no.
**Ground Truth: yes, yes, no**

### CoT Answer by Phi-4

**Steps:**

- Identifies known truth/lie statuses (Jake at airport, Jaxon at amusement park, Devika at observatory).
- Lists statements made by target individuals (Farm, Restaurant, Observatory).
- Correctly deduces Devika (Observatory) lies.
- Correctly deduces Luna (Restaurant) tells the truth because her statement about Devika is true.
- Analyzes Max (Farm): Notes Elowen and Liam say Max lies, but **fails to fully leverage the crucial deduction that Luna (Restaurant) tells the truth**.
- Instead of using Max's statement "The person at the restaurant tells the truth" (which is true, as Luna tells the truth) to prove Max is a truth-teller, it focuses on other statements and concludes Max is "likely lying".

**Flaw:** Missed a key logical step. It correctly deduced Luna at the Restaurant is a truth-teller. Max at the Farm states that the person at the Restaurant tells the truth. Since Max's statement about Luna is true, Max himself must be a truth-teller. The CoT failed to make this final deduction about Max, leading to an incorrect conclusion for the Farm.
**Final Answer: no, yes, no**                                    **Incorrect**

### DyFlow Answer by Phi-4

**Stage 1: Decomposition and Planning**
`DECOMPOSE_PROBLEM, GENERATE_PLAN`
Breaks down the complex puzzle into explicit constraints, person-location mapping, and generates a systematic plan for logical deduction. This ensures all pieces of information are organized for analysis.

**Stage 2: Solution Generation and Review**
`GENERATE_ANSWER, REVIEW_SOLUTION`
Generates a step-by-step solution using the structured information. Crucially, during the deduction phase (as seen in the final organized solution), it correctly identifies:

- Devika (Observatory) lies (known).
- Luna (Restaurant) tells the truth because she correctly states Devika lies.
- **Max (Farm) tells the truth because he states the person at the Restaurant (Luna) tells the truth, and Luna has been correctly deduced to be a truth-teller.** This is the critical deduction missed by CoT.

The `REVIEW_SOLUTION` step confirms the logical consistency and completeness of the deductions against all constraints.

**Stage 3: Finalization**
`ORGANIZE_SOLUTION, TERMINATE`
Organizes the verified, correct solution and presents the final answer in the required format.

**Final Answer: yes, yes, no**                                    **Correct**

## Case Study (LiveBench)

### Problem Description

**Task:** Suppose I have a regular heptagon, and I can make four straight cuts. Each cut cannot pass through any of the vertices of the heptagon. Also, exactly three of the cuts must intersect at a single point within the heptagon. What is the maximum number of resulting pieces? Output answer as a single integer.

**Ground Truth: 10**

### CoT Answer by Phi-4

**Steps:**

- Identifies constraints (4 cuts, no vertices, 3 concurrent).
- Mentions general formula for maximum regions by $n$ lines in a plane: $R(n) = \frac{n(n+1)}{2} + 1$.
- Notes three concurrent lines form 6 regions instead of 7 (correct for 3 lines in a plane).
- Attempts calculation: "Initial regions formed by three intersecting lines: 6". This is likely referring to the plane, not pieces in the heptagon (3 concurrent cuts divide a convex shape into 5 pieces).
- States "The maximum number of regions formed by four lines, with three intersecting at a point, is 11". This is incorrect. The maximum regions in the plane for 4 lines with 3 concurrent is 10 ($R(4) - (3-1)(3-2)/2 = 11 - 1 = 10$). More importantly, it misapplies plane region logic to pieces within a bounded shape under concurrency. A line crossing $k$ existing segments adds $k + 1$ pieces.

**Flaw:** Miscalculated the maximum number of regions in the plane for 4 lines with 3 concurrent (states 11, should be 10). More fundamentally, failed to correctly reason about how cuts add *pieces within the bounded heptagon*, especially under the concurrency constraint. It seems to confuse total regions in the plane with pieces within the shape, and incorrectly handles the impact of the fourth cut.

**Final Answer:** 11         **Incorrect**

### DyFlow Answer by Phi-4

**Stage 1: Decomposition, Solution Generation, and Review**
`DECOMPOSE_PROBLEM, GENERATE_ANSWER, REVIEW_SOLUTION`
DyFlow begins by decomposing the task into fundamental constraints: 4 cuts, 3 of which must intersect at a single point, and none of which may pass through vertices. The planner generates a solution by correctly reasoning that three concurrent cuts form 6 regions inside the heptagon, and the fourth cut, intersecting all three, adds 4 new regions (1 for each segment crossed). This leads to a total of $6 + 4 = 10$ pieces.

The `REVIEW_SOLUTION` operator verifies this reasoning, ensuring that the concurrency constraint and the incremental piece count are correctly applied. It confirms that no logical or constraint violations are present and that the plan is internally consistent.

**Stage 2: Finalization**
`ORGANIZE_SOLUTION, TERMINATE`
The final verified solution is organized for presentation. The planner concludes the task by outputting the answer in the required format.

**Final Answer: 10**         **Correct**

## Case Study (SocialMaze)

### Problem Description

**Task:** Deduce the criminal and Player 1's role in a 6-player social deduction game (3 Investigators, 1 Criminal, 1 Rumormonger, 1 Lunatic) after 3 rounds of statements. Player 1 is told they are the Criminal. Investigators always tell the truth about the Criminal. Rumormongers believe they are Investigators but may be incorrect. Lunatics believe they are Criminals and may be truthful or false. Output format: Final Criminal Is Player [X]. My Role Is [Y or Unknown]. (Only provide part of the problem description due to limited space.)
**Ground Truth:** Final Criminal Is Player 4. My Role Is Lunatic.

### CoT Answer by Phi-4

**Steps:**

- Summarizes rules and statements per round.
- Notes P6 accuses P1 in R1, P5 denies P1 in R1.
- Notes P2,3,4,6 deny P1 in R2/R3 (mostly).
- Notes P2,3 accuse P4 in R3.
- **Flawed Deduction on Player 1:** Concludes Player 1 is "likely the Criminal" because they were told they are the Criminal and their statements are inconsistent. This ignores the rule that Lunatics are *told* they are Criminals and also make inconsistent statements, and more importantly, contradicts the evidence from multiple players denying P1 is the criminal.
- **Flawed Deduction on Player 6:** States P6 is "likely an Investigator, as they consistently accuse Player 1". This is false; P6 accuses P1 (R1), denies P1 (R2), accuses P3 (R3). P6 is *inconsistent*.
- Attempts role assignment based on these flawed deductions, leading to P1 as Criminal and P2,3,6 as Investigators (which contradicts P2,3,4,6 denying P1 in R2/R3).

**Final Judgment:** Final Criminal: 1, My role: Criminal                    **Incorrect**

### DyFlow Answer by Phi-4

**Stage 1: Initial Deduction and Review**
GENERATE_ANSWER - Analyzes the statements and role rules to produce an initial answer: Player 4 is the Criminal; Player 1's role remains uncertain.
REVIEW_SOLUTION - Checks the consistency of the deduction. Confirms that Player 4 is a strong candidate for Criminal but flags the ambiguity in Player 1's role as a minor issue.

**Stage 2: Refinement and Re-Review**
REFINE_ANSWER - Reevaluates Player 1's status using rule-based reasoning: being told they are the Criminal while evidence contradicts that claim aligns with the definition of a Lunatic.
REVIEW_SOLUTION - Validates the refined answer. Confirms that all constraints are satisfied and the role assignment is consistent and complete.

**Stage 3: Finalization**
ORGANIZE_SOLUTION, TERMINATE - Formats and outputs the final answer.

**Key Deductions:**

1. P2 and P3 consistently accuse Player 4. Since Investigators always speak truthfully about the Criminal, this supports P4 as the true Criminal.
2. Player 1 was told they are the Criminal, but all external evidence contradicts this. Under the game rules, this behavior corresponds to the Lunatic role.

**Final Judgment:** Final Criminal Is Player 4. My Role Is Lunatic.                    **Correct**

## Case Study (Code Generation - Simple Parsing)

### Problem Description

**Task:** Write a Python function `fruit_distribution(s, n)` that takes a string `s` ("X apples and Y oranges") and a total fruit count `n`, and returns the number of mangoes (`n - X - Y`). Examples provided.

**Ground Truth:** Code that correctly parses the string `s` to extract X and Y, and returns `n - X - Y`.

### CoT Answer (Original Code) by Phi-4

**Summary:** The generated code extracts the number of apples correctly using `parts[0]`, but incorrectly attempts to extract oranges using `parts[4]`, which results in a `ValueError` when parsing the word `"oranges"`.

**Result:** Code fails at runtime due to incorrect parsing. **Incorrect**

### DyFlow Answer by Phi-4

**Stage 1: Initial Generation & Review**
`GENERATE_CODE` produces a first draft, likely with the same parsing flaw.
`REVIEW_SOLUTION` identifies the issue via failed test cases. (Status: Minor Issues)

**Stage 2: Test Construction & Refinement**
`DEFAULT` - Dynamically constructs targeted test cases to diagnose parsing behavior and edge failures.
`REFINE_CODE` - Uses the generated tests to revise the parsing logic. Attempts an improved implementation.
`REVIEW_SOLUTION` - Reviews the revised code. Some issues remain unresolved in edge cases. (Status: Minor Issues)

**Stage 3: Further Refinement & Final Review**
`REFINE_CODE` - Performs another round of refinement. This time adopts robust parsing (e.g., regex).
`REVIEW_SOLUTION` - Validates that all test cases now pass. (Status: Accept)

**Stage 4: Finalization**
`ORGANIZE_SOLUTION` - Formats and documents the final code.
`TERMINATE` - Outputs the final result and concludes the workflow.

**Result:** Code passes all tests and produces correct output. **Correct**

## G   Prompt Template

**GENERATE_CODE**

You are an expert in solving coding problems. Generate Python code based on the following context and guidance.

Context: {context}
Guidance: {guidance}

Your code must:
1. Define a function named `solve` that calculates and returns the final result.
2. Clearly comment each computational step.
3. Obtain necessary inputs from within the function or global variables (no function parameters).

Output Format:
```python
# Your generated code here (use the main function name 'solve')
```

---

**GENERATE_ANSWER**

You are an expert in solving reasoning problems. Think step by step to solve the problem using the context and guidance.

Context: {context}
Guidance: {guidance}

Output Format:
Reasoning: <You should think step by step to solve the problem.>
Answer:

---

**REVIEW_SOLUTION**

You are a careful reviewer trained to detect logical and mathematical errors. Your job is to critically evaluate the solution for correctness, soundness, and completeness.

Context: {context}
Guidance: {guidance}

Instructions:
- Try to find mistakes at every step of the given answer.
- Bring the answer back to the original question and check if there is anything that does not meet the requirements.

Output Format:
Review Details: <step-by-step review>
Overall Verdict: <accept/minor_issues/major_issues/reject>

## DECOMPOSE_PROBLEM

You are an expert in decomposing problems. Break down the original problem into clearly defined, structured sub-tasks.

Context: {context}
Guidance: {guidance}

Instructions:
- Clearly outline each distinct sub-task.
- Do not attempt to solve any sub-task.
- Maintain logical completeness.
- Decompose the problem into 2–4 steps at most.

Output:
<your_decomposed_problem>

## GENERATE_PLAN

You are an expert in generating step-by-step executable plans. Generate a step-by-step executable plan to approach the given problem.

Context: {context}
Guidance: {guidance}

Instructions:
- Clearly number each step.
- Ensure each step is actionable and logically sequenced.
- Do not solve the problem here, only provide the plan.
- Give 2–4 steps at most.

Output Format:
Solution Plan:
<step_id>: <description>
<step_id>: <description>

## REFINE_CODE

You are an expert in refining Python code. Refine the existing Python code based on context and guidance.

Context: {context}
Guidance: {guidance}

Instructions:
- Correct errors or inefficiencies identified.
- Clearly comment important logic or corrections.
- Maintain the main function name as `solve`.

Output Format:
```python
# Your refined code here (use the main function name 'solve')
```

## REFINE_ANSWER

You are an expert in refining answers. Refine the existing answer based on context and guidance.

Context: {context}
Guidance: {guidance}

Output Format:
Answer: <your refined answer>

## ORGANIZE_SOLUTION

You are an expert in organizing solutions. Clearly organize the final solution for presentation based on the provided context and guidance.

Context: {context}
Guidance: {guidance}

Instructions:
- Clearly present final reasoning steps and results.
- Ensure alignment with the problem's required formatting.
- Omit irrelevant or incorrect previous attempts.

Output:
<your_organized_solution>

## ENSEMBLE

You are an expert in generating multiple valid and diverse solutions using distinct logical approaches.

Context: {context}
Guidance: {guidance}

Instructions:
- Each solution must independently satisfy all constraints.
- Clearly separate each distinct reasoning path and solution.

Output:
<your_ensemble_output>

## DEFAULT

You are an expert in executing actions strictly according to the given context and guidance.

Context: {context}
Guidance: {guidance}

Instructions:
- Follow every detail of the instructions carefully.
- Ensure output exactly matches the requested format.

Output:
<your_output>

