# OpenReview forum: "DyFlow: Dynamic Workflow Framework for Agentic Reasoning"
_NeurIPS.cc/2025/Conference — NeurIPS 2025 poster_

### Official Review · Reviewer_w8CR · 2025-06-11

**Clarity:** 3
**Significance:** 3
**Originality:** 3
**Rating:** 4
**Confidence:** 4

**Summary:**

This paper introduces DyFlow, a dynamic workflow framework for LLM-based agents that adaptively constructs and updates subgoal plans during execution. It addresses a key limitation in existing agent frameworks—namely, their reliance on static, predefined workflows that cannot respond effectively to intermediate execution failures or evolving task contexts.

DyFlow features a two-tier architecture:
- A Designer, which generates structured stage subgraphs (subgoal plans) based on the current task state
- An Executor, which runs the subgraphs using parameterized operator templates

The Designer is trained using a two-phase strategy:
- Supervised Fine-Tuning (SFT) on high-quality subgraphs, followed by
- KL-regularized Trajectory Optimization (KTO), where the model learns from its own successes and failures.

The system is evaluated across five reasoning domains—math, code, medical, logic, and social—showing strong gains in accuracy, Pass@k stability, and cross-task generalization. Ablation studies and case analyses show the importance of dynamic planning and the self-play optimization loop.

**Questions:**

### Questions

- How does DyFlow differ empirically and conceptually from OctoTools, which also supports dynamic, feedback-aware planning through a planner–executor loop?
- Why does PubMedQA performance improve in the cross-task generalization setting (75.20 in Table 3) compared to the task-specific training setting (72.91 in Table 1)?

**Ethical Concerns:**

["NO or VERY MINOR ethics concerns only"]

**Final Justification:**

The authors have adequately addressed the majority of my concerns in the rebuttal. However, the lack of direct comparison with OctoTools—a closely related, training-free framework with overlapping domains—remains a gap.

Overall, while I see clear strengths in the submission, especially in its practical relevance, the remaining issues justify maintaining my original score.

**Limitations:**

The paper includes only a brief mention of future work. It would benefit from a more thorough discussion of its limitations, a clearer positioning of its contributions in relation to recent work, and a more detailed exploration of future research opportunities.

**Quality:**

3

**Strengths And Weaknesses:**

### Strengths

- Clear Motivation & Scope: Addresses a key limitation in LLM agents—static workflows—with a modular, dynamic planning architecture.
- Clean Design: The planner–executor–memory framework is well-structured, extensible, and easy to interpret.
- Strong Empirical Results: Demonstrates consistent gains across diverse reasoning domains, with good ablations and generalization tests.
- Open-Model Compatibility: Shows that dynamic planning and planner refinement work well even with lightweight open models like Phi-4, supporting practical deployment.

### Weaknesses

- Limited Novelty in Dynamic Planning: The planner–executor loop closely resembles recent frameworks like OctoTools, which also support step-wise, feedback-based replanning. The paper lacks a detailed conceptual or empirical comparison to such systems.
- Weak Medical Evaluation (Table 1): PubMedQA is not an ideal benchmark for medical reasoning due to its limited diagnostic complexity and constrained answer format (yes/no/maybe). The paper does not evaluate on other more common medical reasoning datasets (e.g., MedQA, MedMCQA, MedXpert).
- Missing Baselines (Tables 1 & 2): The paper omits a key SFT-only baseline (e.g., Phi-4 with simple question-answer SFT), which is needed to isolate the effect of planning and complicated agentic framework. Additionally, Table 2 does not include a direct baseline using the untrained Phi-4 model, unlike GPT and Claude.
- Incomplete Executor Analysis (Table 4): All evaluations use Phi-4 as the executor. It remains unclear whether DyFlow still provides meaningful gains with stronger executors like GPT-4o or Claude-3.

---

> ### Author Rebuttal · Authors · 2025-07-30
>
> We sincerely thank the reviewer for the detailed and thoughtful comments. We have carefully revised the paper based on your suggestions, and we address each point below.
>
> ---
>
> **$Q_1:$ Limited Novelty in Dynamic Planning: The planner–executor loop closely resembles recent frameworks like OctoTools, which also support step-wise, feedback-based replanning. The paper lacks a detailed conceptual or empirical comparison to such systems.**
>
> $A_1:$ We thank the reviewer for drawing a connection to OctoTools. While both DyFlow and OctoTools adopt planner–executor paradigms with feedback, their **core design goals and planning mechanisms differ substantially**.
>
> First, the target domains differ: OctoTools is designed for multimodal, tool-augmented reasoning (e.g., vision combined with API queries), whereas DyFlow focuses on symbolic and numerical reasoning tasks such as math, coding, logic, and medical QA, where dynamic workflow restructuring is central to performance.
>
> Second, the planning granularity differs: OctoTools selects one tool at a time for each step, while DyFlow’s Designer generates an entire subgraph of symbolic operators per stage and re-plans after execution. This enables more structured adaptations, such as inserting or retracting subgoals when needed.
>
> Third, DyFlow trains its planner explicitly, while OctoTools relies on a zero-shot, frozen planner. Our Designer undergoes two training phases, which enable it to internalize complex preferences and learn robust control behaviors.
>
> We appreciate the reviewer’s suggestion and will add a discussion section to clarify these distinctions. We also view OctoTools as an elegant and impactful system, and we are excited to explore how some of its design principles (such as flexible tool interfacing) can inform future iterations of DyFlow.
>
> **$Q_2:$ Weak Medical Evaluation (Table 1): PubMedQA is not an ideal benchmark for medical reasoning due to its limited diagnostic complexity and constrained answer format (yes/no/maybe). The paper does not evaluate on other more common medical reasoning datasets.**
>
> $A_2:$ We appreciate the reviewer’s thoughtful feedback. Our primary goal in selecting PubMedQA was not to fully capture the complexity of clinical diagnosis, but to demonstrate DyFlow’s ability to generalize across **diverse reasoning domains**, including biomedical literature, within a unified planning framework.
>
> PubMedQA offers a controlled, domain-specific benchmark that still requires models to extract and verify factual claims from noisy, technical abstracts. While the answer space is limited, it provides a useful proxy for evaluating dynamic planning under high-stakes, text-intensive conditions. Notably, recent works such as [1] and [2] have continued to adopt PubMedQA due to its relevance and robustness.
>
> That said, we agree that more comprehensive benchmarks like MedQA, MedMCQA, or MedXpert offer deeper diagnostic and multi-hop challenges. Extending DyFlow to these richer settings is a promising direction for future work.
>
> [1] Towards Medical Complex Reasoning with LLMs through Medical Verifiable Problems
>
> [2] EvoBench: Towards Real-world LLM-Generated Text Detection Benchmarking for Evolving Large Language Models
>
> **$Q_3:$ Missing Baselines (Tables 1 & 2): The paper omits a key SFT-only baseline (e.g., Phi-4 with simple question–answer SFT), which is needed to isolate the effect of planning and complicated agentic framework. Additionally, Table 2 does not include a direct baseline using the untrained Phi-4 model, unlike GPT and Claude.**
>
> $A_3:$ We thank the reviewer for highlighting this important baseline. To isolate the effect of dynamic planning, we added a Phi-4 baseline trained purely with SFT on GPT-4.1-labeled question–answer pairs, without any workflow structure or subgoal decomposition. As shown below, this SFT-only model performs competitively, but **DyFlow still consistently outperforms it** across all domains:
>
> | Method        | SocialMaze | PubMedQA | MATH  | LiveBench | HumanEval |
> | ------------- | ---------- | -------- | ----- | --------- | --------- |
> | Phi-4 (SFT)   | 11.45      | 67.73    | 69.60 | 40.67     | 88.07     |
> | DyFlow (Ours) | 17.18      | 72.91    | 76.40 | 48.67     | 92.07     |
>
> These results demonstrate that DyFlow’s improvement comes from its **structured, multi-stage planning**.
>
> Regarding the reviewer’s suggestion to include a zero-shot Phi-4 baseline: in our pilot experiments, an untrained Designer using raw Phi-4 generated valid subgraphs in fewer than 5% of the cases, primarily due to Phi-4’s limited capability. As such, it could not serve as a meaningful or informative baseline. We will clarify this rationale in the revised version.
>
> **$Q_4:$ Incomplete Executor Analysis (Table 4): All evaluations use Phi-4 as the executor. It remains unclear whether DyFlow still provides meaningful gains with stronger executors like GPT-4o or Claude-3.**
>
> $A_4:$ We thank the reviewer for this insightful question. Yes, we have conducted extensive cross-executor evaluations to test DyFlow’s generality beyond Phi-4. As shown in **Figure 4** and detailed in **Appendix D.5 (Table 10)**, the Designer trained with DyFlow self-play transfers seamlessly to stronger executors such as GPT-4o-mini and GPT-4.1-mini, without any additional tuning. In all settings, DyFlow consistently outperforms CoT-style prompting.
>
> This ability to generalize arises from DyFlow’s **decoupled architecture**: the Designer is trained to orchestrate workflows and manage information flow, while treating the Executor as a black-box reasoning module. Since the Designer itself does not perform the reasoning, it can easily interface with a wide range of LLMs, enabling plug-and-play compatibility and strong zero-shot transfer.
>
> **$Q_5:$ Why does PubMedQA performance improve in the cross-task generalization setting (75.20 in Table 3) compared to the task-specific training setting (72.91 in Table 1)?**
>
> $A_5:$ We thank the reviewer for raising this point and sincerely apologize for the confusion caused by the use of abbreviations in Table 3. Specifically, the row labeled **QR corresponds to PubMedQA** (question answering), not MR, which stands for MATH (mathematical reasoning). The full mapping is as follows:
>
> - SR: SocialMaze (social reasoning)
> - CR: HumanEval (code reasoning)
> - MR: MATH (mathematical reasoning)
> - **QR: PubMedQA (medical Question answering)**
> - LR: LiveBench (logical reasoning)
>
> Accordingly, the correct cross-task performance for PubMedQA is **72.11**, as shown in the QR row, which aligns closely with the task-specific performance of 72.91 in Table 1. We will revise the table in the final version by adding a clear legend with these task categories to prevent misunderstanding.
>
> ---
>
> Thank you again for your valuable time and suggestions and we would greatly appreciate your kind support during the discussion phase.

---

> > ### Comment · Reviewer_w8CR · 2025-08-05
> >
> > Thank you, Authors, for addressing my questions and comments. Most of my concerns have been resolved.
> >
> > One note I would like to add regarding the comparison between DyFlow and OctoTools: I recommend including OctoTools as a baseline in future revisions. There are notable similarities between the two frameworks, with OctoTools having the advantage of being training-free. Additionally, OctoTools overlaps in domain coverage with DyFlow, as it has been evaluated on datasets such as MathVista and MedQA.
> >
> > That said, I will maintain my current score. My overall assessment of the paper remains more positive than negative.

---

> > > ### Author Response · Authors · 2025-08-07
> > >
> > > We are truly grateful for your time and thoughtful evaluation! Your feedback has been invaluable in helping us improve the paper.
> > >
> > > Thank you especially for highlighting the importance of a direct empirical comparison with OctoTools. We appreciate the insight that, despite differing design choices, the overlap in domains and the training-free nature of OctoTools make it a meaningful baseline. We will make sure to include this comparison in future versions of the work.

---

### Official Review · Reviewer_FEBY · 2025-06-23

**Clarity:** 3
**Significance:** 4
**Originality:** 3
**Rating:** 5
**Confidence:** 4

**Summary:**

This paper presents a novel dynamic agentic workflow generation framework, named DyFlow, designed for reasoning tasks. It consists of two core components: a designer to decompose the task into sub-goals and refine the plan generation given the intermediate output and feedback; an executor to execute each operation initialized from the operator templates. The designer is trained using a two-phase learning strategy with knowledge distillation and self-play preference optimization. The proposed framework outperforms the existing baselines on different reasoning benchmark datasets.

**Questions:**

* Could the authors explain the motivation that only the designer is trained in the proposed framework? Why is the executor excluded from training? And have the authors considered an end-to-end training including both the designer and executor?
* In Figure 2, could the authors annotate where the executor is in the diagram? The current diagram only includes a designer component.
* In Algorithm 1, could the authors explain what the content is in the memory? What is a list of keys? And how are the inputs retrieved from the memory?

**Ethical Concerns:**

["NO or VERY MINOR ethics concerns only"]

**Final Justification:**

This paper introduces the designer-executor architecture to dynamically decompose the tasks into sub-tasks. Although the entire architecture is not new, it introduces a novel two-stage training strategy to train the designer and executor executes a set of high-level operators to execute the task. I believe this is an interesting work that could provide insights on the dynamic agentic workflow generation.

**Limitations:**

Yes.

**Paper Formatting Concerns:**

No.

**Quality:**

4

**Strengths And Weaknesses:**

**Strengths**
* This paper proposes a novel framework built on a hierarchical designer-executor architecture that dynamically decomposes the task into sub-goals instead of relying on a static workflow.
* The proposed framework DyFlow uses the feedback and outputs from the executor to revise the plan, retry the sub-goal to support fine-grained execution corrections.
* The designer is trained using a novel two-phase learning strategy with knowledge distillation and self-play preference optimization without relying on human-annotated labels.
* DyFlow achieves consistent and significant improvements over the existing baselines across multiple reasoning tasks.

**Weakness**
* The designer instantiates the operators from a finite set of operator templates to construct the workflow graph, which only consists of symbolic decomposition and code execution. The executor doesn't support tools or APIs for more complex reasoning tasks.
* The designer constructs the sub-graph given the task and feedback; however, it is not guaranteed that the constructed sub-graph is valid and optimal for the task. For example, the designer might generate redundant and inefficient workflows.
* The proposed framework needs to use another LLM to generate a condensed state summary to determine the subgoal and a stage subgraph. During the summarization, it is not guaranteed that no information will be lost.

---

> ### Author Rebuttal · Authors · 2025-07-30
>
> We sincerely thank the reviewer for the detailed and thoughtful comments. We have carefully revised the paper based on your suggestions, and we address each point below.
>
> ---
>
> **$Q_1:$ The designer instantiates the operators from a finite set of operator templates to construct the workflow graph, which only consists of symbolic decomposition and code execution. The executor doesn't support tools or APIs for more complex reasoning tasks.**
>
> $A_1:$ We appreciate the reviewer for pointing this out. We agree with the reviewer that DyFlow, in its current form, operates over a **symbolic and code-centric action space**, and does not yet interface with tools such as search engines, databases, or APIs. This was an intentional scope choice: our focus in this work was to test whether dynamic re-planning mechanisms alone, in the absence of external resources, could bring significant gains in reasoning performance. As shown in our experiments (e.g., Table 1), this setup already reveals strong benefits over static or prompt-based workflows.
>
> That said, we recognize that enabling **richer tool integration** and expanding beyond single-modality is essential for building more capable agentic systems. Incorporating such functionalities would involve not only extending the operator template space, but also developing mechanisms for handling noisy or partial tool outputs, and possibly adopting finer-grained credit assignment during training. We will include this point clearly in the revised *Limitations* section and consider it a core direction for future research.
>
> **$Q_2:$ The designer constructs the sub-graph given the task and feedback; however, it is not guaranteed that the constructed sub-graph is valid and optimal for the task. For example, the designer might generate redundant and inefficient workflows.**
>
> $A_2:$ We thank the reviewer for raising this insightful point. It is true that DyFlow does not guarantee the generated subgraphs are **globally optimal or minimal** in size. However, this is an intentional aspect of our learning-based design. Our goal is to train the Designer to produce **empirically effective subgraphs**—those that work well in practice, even if not theoretically minimal.
>
> To this end, our two-stage training procedure plays a crucial role. The SFT phase provides strong initial patterns, while the KTO phase encourages the Designer to favor subgraphs that lead to success and penalizes those that result in failures or redundant reasoning. This allows the model to iteratively refine its planning behavior based on trajectory-level feedback.
>
> Although optimality is not explicitly enforced, the consistent improvements across tasks and benchmarks, as shown in Table 1 and Table 3, demonstrate that the Designer learns to generate workflows that are both effective and efficient for the reasoning problems at hand.
>
> **$Q_3:$ The proposed framework needs to use another LLM to generate a condensed state summary to determine the subgoal and a stage subgraph. During the summarization, it is not guaranteed that no information will be lost.**
>
> $A_3:$ We appreciate the reviewer for raising this point. In DyFlow, the summarizer serves a **well-scoped and lightweight** role: it condenses intermediate outputs and operator results into a structured state for the Designer to process. It does not perform complex reasoning or decision-making. This task is effectively a summarization problem, which current LLMs handle with high reliability.
>
> In our experiments, we observed very few instances where the summarizer omitted critical information. Furthermore, DyFlow’s **multi-stage planning architecture** enables self-correction: even if a salient detail is missed in one stage, it can often be recovered or revised in subsequent ones. In practice, we found that summarization imperfections rarely impact the final outcome, thanks to this built-in robustness.
>
> **$Q_4:$ Could the authors explain the motivation that only the designer is trained in the proposed framework? Why is the executor excluded from training? And have the authors considered an end-to-end training including both the designer and executor?**
>
> $A_4:$ We thank the reviewer for this insightful question. The core motivation for training only the Designer lies in the **modular nature** of our framework. In DyFlow, the Designer is responsible for planning and organizing the reasoning workflow, while the Executor focuses on carrying out operator-level actions based on those plans. Since the Executor is treated as a black-box solver, we do not require it to be trained jointly with the planner.
>
> This separation brings several practical benefits. First, it allows the trained Designer to transfer seamlessly across different Executors without any additional tuning. As shown in Figure 4 and Appendix D.5 (Table 10), a Designer trained via DyFlow self-play consistently improves performance when paired with new executors such as GPT-4o-mini or GPT-4.1-mini. Second, avoiding fine-tuning of the Executor significantly reduces training cost, while maintaining compatibility with off-the-shelf LLMs. Finally, this architectural modularity enables clearer debugging, reuse, and **system interpretability**—advantages that would be compromised in an end-to-end setting.
>
> While it is possible in principle to jointly train both components, doing so would increase optimization complexity and compromise the generality of the approach. Our results demonstrate that learning to plan is sufficient to achieve strong performance across a wide range of tasks.
>
> **$Q_5:$ In Figure 2, could the authors annotate where the executor is in the diagram? The current diagram only includes a designer component.**
>
> $A_5:$ We thank the reviewer for pointing this out. In our current diagram, the Executor is **implicitly represented** within each instantiated operator in the subgraph. Specifically, each operator node (e.g., Op Instance) corresponds to a call to the Executor, which performs reasoning or code execution based on the given operator definition and input.
>
> To keep the figure concise, we abstracted away the Executor as part of the operator execution process. We agree that **explicitly annotating** the Executor component would improve clarity, and we will revise the figure accordingly in the final version to make this architectural separation more transparent.
>
> **$Q_6:$ In Algorithm 1, could the authors explain what the content is in the memory? What is a list of keys? And how are the inputs retrieved from the memory?**
>
> $A_6:$ We thank the reviewer for this detailed question. In DyFlow, the memory $\mathcal{M}$ is implemented as a **global key–value dictionary** that records all intermediate outputs produced by operator executions throughout a session. Each result is written back into the memory under a unique key—typically based on the operator’s ID. This enables downstream operators to reuse prior outputs without recomputation and allows the Designer to access the full task history when replanning.
>
> In each operator instance $(O_k, \phi, \psi)$, the list $\psi$ denotes the **ordered list of memory keys** whose values serve as inputs to that operator. During execution, the Executor retrieves inputs via simple dictionary lookups: for each key $k \in \psi$, it queries $\mathcal{M}[k]$, constructs the input, and applies the operator logic accordingly. This memory-based design ensures stateless, traceable execution and transparent data flow across reasoning stages. We will clarify this point in the revised version of the paper.
>
> ---
>
> Thank you again for your valuable time and suggestions and we would greatly appreciate your kind support during the discussion phase.

---

> > ### Comment · Reviewer_FEBY · 2025-08-05
> > **Response to authors**
> >
> > Thank you for the clarifications!
> >
> > I have a follow-up question regarding the set of operators. Could the authors comment on the scalability of operators? As mentioned previously, the current set of operators cover a symbolic and code-centric action space. How does it scale when we need to integrate with more tools such as search engine, databases or APIs? Do we have to manually design more operators when we need to enable a large number of tools?

---

> > > ### Author Response · Authors · 2025-08-06
> > >
> > > Thank you for the follow-up question and for your continued engagement.
> > >
> > > We agree that the scalability of the operator set is a crucial consideration for extending DyFlow to more complex and tool-intensive scenarios. Our current operator library focuses on symbolic and code-centric reasoning, but the DyFlow framework is designed to be scalable—both in terms of operator composition and tool integration.
> > >
> > > To support the incorporation of tools such as search engines, databases, or APIs, we envision introducing a generalized TOOL_EXECUTION operator. This operator encapsulates the logic for interacting with external tools while remaining agnostic to the specific tool type. In practice, tools can register themselves using a structured interface (e.g., via the MCP protocol), specifying their input/output signatures and schema. The Executor can then dynamically format and route calls to the appropriate tool at runtime. This design decouples tool selection from flow planning: the Designer only needs to decide when to invoke TOOL_EXECUTION, while the Executor handles tool selection and execution based on the registry.
> > >
> > > During training, the Designer can be exposed to a small number of representative tool use cases via system prompts and minimal examples. Crucially, the goal is not to imitate specific tool behaviors, but to learn when tool invocation is warranted given the context. This facilitates generalization to previously unseen tools in a zero-shot setting, leveraging DyFlow’s modular and extensible architecture.
> > >
> > > We appreciate this suggestion and agree that robust tool integration is an important future direction. We are actively working toward extending DyFlow in this way and will incorporate this line of development into future iterations of the framework.

---

> > > > ### Comment · Reviewer_FEBY · 2025-08-06
> > > > **Response to authors**
> > > >
> > > > Thank you for the clarifications! Using an abstract TOOL_EXECUTION operator to make it agnostic to specific tool types makes sense to me. Definitely would be interested in seeing how DyFlow scales to more tools so that it can handle more complex tasks reliably.
> > > >
> > > > I will retain my overall score as it is already positive.

---

> > > > > ### Author Response · Authors · 2025-08-07
> > > > >
> > > > > Thank you for your kind support and thoughtful feedback. We're glad our design direction resonated with you, and we look forward to scaling DyFlow to support more complex tool use. We truly appreciate your engagement throughout the review process.

---

### Official Review · Reviewer_y5j8 · 2025-07-01

**Clarity:** 4
**Significance:** 4
**Originality:** 4
**Rating:** 5
**Confidence:** 4

**Summary:**

This paper introduces a designer-executor framework for agentic workflow that performs dynamic reorganization of task subgoals to overcome the issues with static or rule-based based agentic workflows. The designer agent observes the current task state and plans the next step of the reasoning process. This plan occurs in the form of subgraph of ops. The executor model then executes the operations in this subgraph. The proposed designer, the main contribution of this work, is trained using a combination of policy distillation and self-play to significantly improve its design capabilities.  The feedback framework allows the designer to rethink the workflow steps in a task to avoid terminating errors as an output. They show that DyFlow is able to perform much better than prompt-based methods and automated agent frameworks for different reasoning domains, one-shot generalization across other held-out domains, better evaluation on coding benchmarks, robust across different designer models. They also highlight the importance of the dynamic planning agent using the ablation, where they remove dynamic planning and only generate a static workflow.

**Questions:**

My questions are bundled with the limitations stated above.

**Ethical Concerns:**

["NO or VERY MINOR ethics concerns only"]

**Final Justification:**

I am maintaining my final score for the paper. I am satisfied with the existing rebuttal from the authors. The work, is a novel contribution in designer-executor framework for agentic workflow and having the interesting discussion especially from the other reviewer W8Cr, I am going to maintain my current rating

**Limitations:**

Please see weaknesses section

**Quality:**

4

**Strengths And Weaknesses:**

+ The idea of moving from one-shot workflow generation to feedback-driven subgoal re-planning paradigm is highly significant
+ The authors provide rigorous benchmarking across multiple reasoning domains
+ It appears that the learned agents achieve true generalization based on the cross task results in Table 3
+ SFT + KTO seems to work very well on an open source model compared to propietary models
+ The ablation clearly shows the importance of the planner as an important contribution of this work

- How does the adaptive workflow affect latency or, equivalently, real time nature of the response or speed of task execution. Is there a trade-off study of speed of completion vs quality of task execution?
- How vital is the diversity of the operator templates available for the work?

---

> ### Author Rebuttal · Authors · 2025-07-30
>
> We sincerely thank the reviewer for the detailed and thoughtful comments. We have carefully revised the paper based on your suggestions, and we address each point below.
>
> ---
>
> **$Q_1:$ How does the adaptive workflow affect latency or, equivalently, real time nature of the response or speed of task execution? Is there a trade-off study of speed of completion vs quality of task execution?**
>
> $A_1:$ We thank the reviewer for raising this important point. As detailed in Appendix D.4 and Table 8, we provide a full analysis of DyFlow’s pipeline cost, including token usage from both the Designer and Executor during inference on the MATH benchmark.
>
> While DyFlow introduces modest computational overhead due to its multi-stage planning structure, the total token cost remains within a reasonable range. As shown in Table 8, DyFlow consumes slightly more tokens than MaAS or AFlow (approximately **1.4×–3×**), but achieves a **2.0–2.8% improvement** in accuracy. This suggests that DyFlow’s adaptive workflow introduces a moderate but controlled cost, which is justified in settings where reasoning accuracy is prioritized.
>
> Moreover, DyFlow remains much more efficient than prompt-based agents like LLM-Debate and Self-Refine, which incur substantially higher inference costs without dynamic planning. Overall, DyFlow offers a balanced trade-off between efficiency and planning quality, particularly in tasks that benefit from structured reasoning.
>
> **$Q_2:$ How vital is the diversity of the operator templates available for the work?**
>
> $A_2:$ We appreciate the reviewer’s question. Empirically, we find that the diversity of operator templates plays a crucial role in DyFlow’s performance. While the core operators (e.g., `DECOMPOSE_PROBLEM`, `GENERATE_ANSWER`, `TERMINATE`) handle basic reasoning, many tasks benefit significantly from higher-level planning and refinement operations.
>
> To verify this, we conducted an ablation study comparing the full DyFlow system against a reduced variant using only five core templates. The simplified version omits templates responsible for iterative optimization, plan construction, and ensemble voting. Evaluation on 100 random examples per task yields the following results:
>
> | Task       | Full | Core | Δ    |
> | ---------- | ---- | ---- | ---- |
> | SocialMaze | 17%  | 12%  | −5%  |
> | PubMedQA   | 71%  | 70%  | −1%  |
> | MATH       | 75%  | 71%  | −4%  |
> | LiveBench  | 48%  | 44%  | −4%  |
> | HumanEval  | 93%  | 90%  | −3%  |
>
> These drops highlight that template diversity is not merely auxiliary—it **directly contributes** to DyFlow’s ability to correct mistakes, explore alternatives, and handle structurally diverse tasks. In particular, tasks like SocialMaze and LiveBench, which require multi-step re-evaluation and flexible planning, are disproportionately affected by the removal of these advanced operators.
>
> While this study focuses on hand-designed templates, we agree that learning or inducing new operator templates dynamically is an important direction. We will include this explicitly in the revised *Limitations* section as a target for future research.
>
> ---
>
> Thank you again for your valuable time and suggestions and we would greatly appreciate your kind support during the discussion phase.

---

> > ### Comment · Reviewer_y5j8 · 2025-08-04
> >
> > Thank you for the thorough rebuttal. The diversity of the operators does seem important! I am going to retain the current overall score for the paper.

---

> > > ### Author Response · Authors · 2025-08-07
> > >
> > > We are truly grateful for your time and evaluation! Your feedback is invaluable, and we will certainly incorporate your suggestions into the final version of the paper. Thank you again for your thoughtful engagement.

---

### Official Review · Reviewer_E5YQ · 2025-07-03

**Clarity:** 3
**Significance:** 3
**Originality:** 3
**Rating:** 4
**Confidence:** 4

**Summary:**

The paper introduces DyFlow, a two-level framework that allows large-language-model (LLM) agents to build and revise reasoning workflows on the fly. A high-level Designer turns the current task state into a small “stage sub-graph” that specifies (i) a sub-goal and (ii) a sequence of operator instances (e.g. DECOMPOSE_PROBLEM, GENERATE_ANSWER, REVIEW_SOLUTION, TERMINATE). A low-level Executor (any LLM) runs those operators, pushing intermediate outputs into a shared memory. After each stage, the updated state—including tool outputs, error flags and prior plans—is summarised and fed back to the Designer, which may keep, refine, back-track, or abandon the plan.

**Questions:**

1. How does performance degrade when the operator set is reduced or automatically expanded?
2. What fraction of tasks require a second or third re-planning step, and how often does re-planning fix an error vs cause thrashing?
3. Can the Designer trained under DyFlow self-play transfer to an executor of a different architecture (e.g., Mistral-7B) with zero additional tuning?
4. What are the worst observed failures (e.g., infinite loops, irrelevant operator sequences) and how are they detected or mitigated?
5. Could DyFlow be integrated with retrieval-augmented tool calls by treating search as an operator, and if so, how would the preference signal distinguish relevant from spurious retrievals?

**Ethical Concerns:**

["NO or VERY MINOR ethics concerns only"]

**Final Justification:**

The authors provided comprehensive responses that addressed most of my technical concerns:

Scope Clarification: The intentional focus on symbolic/numerical reasoning rather than tool-augmented tasks was well-justified, demonstrating that dynamic re-planning alone yields significant benefits.
Safety and Code Generation: Quantitative evidence showed 99.4% execution success rate (163/164 HumanEval tasks) with proper sandboxing, addressing my concerns about code safety and reliability.
Operator Set Analysis: The ablation study comparing full vs. core operators (5-template subset) effectively demonstrated the importance of operator diversity, with consistent 3-5% performance drops across domains when higher-level operators were removed.
Re-planning Effectiveness: Analysis of 100 SocialMaze trajectories showed that re-planning successfully corrects errors (9 cases) with minimal thrashing (2 cases), validating the feedback mechanism.
Cross-Architecture Transfer: Confirmation that the Designer transfers to different executors (GPT-4o-mini, etc.) without additional tuning demonstrates the framework's modularity and practical value.

Remaining Limitations:

Domain Scope: Evaluation remains limited to single-modal textual reasoning tasks, lacking validation on multimodal or tool-augmented scenarios that are central to modern agentic systems.
Theoretical Foundations: While practical convergence was demonstrated, formal guarantees about planning error bounds and sample complexity remain absent.
Hyperparameter Sensitivity: Critical settings (temperature 0.01, step cap) were set ad hoc without sensitivity analysis, limiting understanding of robustness.

Weight Assignment:
I assign high weight to the technical innovation in dynamic workflow generation and the strong empirical results across five diverse domains, including impressive zero-shot generalization (SocialMaze: 17% vs 6% CoT baseline). The cost efficiency (≤2% of proprietary model costs while achieving comparable performance) represents significant practical value.
However, I assign moderate weight to the limited scope concerns, as the restriction to symbolic reasoning limits immediate applicability to broader agentic AI scenarios requiring tool integration.

**Limitations:**

The authors note the absence of external tools; I would add (i) potential brittleness to summariser quality, if fsummary misses salient errors, the Designer cannot fix them, and (ii) difficulty in scaling preference labels beyond binary success/failure, which may under-train fine-grained planning choices.

**Quality:**

3

**Strengths And Weaknesses:**

Strengths
1. The Designer is queried every stage, letting DyFlow abandon or rewrite entire sub-goal trees when intermediate checks fail—something one-shot planners and static graphs cannot do.
2. Training a 14 B open-weight Phi-4 with SFT + KTO yields performance on par with 70–>100 B proprietary models, at ≤ 2 % of their dollar cost (Table 9).
3. Five domains, including two that are completely unseen during training, demonstrate cross-task generalisation. DyFlow is zero-shot on SocialMaze yet triples CoT accuracy (17 % vs 6 %).
4. Eleven operator templates (generate, decompose, review, refine, code, ensemble, terminate, …) achieve good coverage without exploding search space; Figure 6 shows domain-specific usage patterns.
5. The authors avoid unstable RL and noisy per-step judges; instead they use trajectory-level success/failure, which yields a tractable, low-variance preference loss.
6. Removing dynamic planning, dynamic operators, SFT or KTO each harms performance, isolating where the gains come from (Table 4).
7. Token-level budget for Designer vs Executor and for training vs inference lets practitioners reproduce or adapt the system.

Weaknesses
1. All tasks are single-modal textual problems requiring symbolic or numerical reasoning; no experiments involve web search, database queries, images, or embodied actions—yet tool-use is a selling point of agentic frameworks.
2. The preference phase is fully offline. The system does not adjust the Designer on-the-fly when facing new domains (e.g. SocialMaze), missing a chance to demonstrate continual improvement.
3. The paper proves DyFlow is “never worse” than a static policy by inclusion, but supplies no bound on performance loss due to planning error, nor on sample complexity of the KTO phase.
4. Safety claims (e.g. all generated code runs, MI > 40) are anecdotal; there is no quantitative report of crashes, hallucinated file paths, or malicious prompt injections.
5. Adding new capabilities still requires designing templates and writing prompt instructions; the paper’s “dynamic operator” claim is limited to parameter instantiation, not operator induction.
6. Critical settings (planner temperature 0.01, step cap T_max) are fixed ad hoc. Sensitivity plots would clarify robustness.

For more detail:
1. Incorporating environments such as HotpotQA (search + multi-hop) or TabMWP (table look-ups) would test DyFlow’s operator abstraction and executor interface.
2. A small-variance, in-run critic (even heuristic) could supply incremental rewards and perhaps close the remaining gap to SOTA GPT-4-based solvers.
3. The memory dictionary M is key to DyFlow; exploring keyed vs vector memory, or pruning strategies, could reveal scalability ceilings.
4. Dynamic, self-revising agents might mask provenance or increase automation risks; the paper should add a short ethics or safety outlook.

---

> ### Author Rebuttal · Authors · 2025-07-30
>
> We sincerely thank the reviewer for the detailed and thoughtful comments. We have carefully revised the paper based on your suggestions, and we address each point below.
>
> ---
>
> **$Q_1:$ All tasks are single-modal textual problems requiring symbolic or numerical reasoning**
>
> $A_1:$​ We agree with the reviewer that DyFlow, in its current form, operates over a symbolic and code-centric action space, and does not yet interface with tools such as search engines, databases, or APIs. This was an intentional scope choice: *our focus in this work was to test whether dynamic re-planning mechanisms alone, in the absence of external resources, could bring significant gains in reasoning performance*. As shown in our experiments (e.g., Table 1), this setup already reveals strong benefits over static or prompt-based workflows.
>
> That said, we recognize that enabling richer tool integration and expanding beyond single-modality is essential for building more capable agentic systems. Incorporating such functionalities would involve not only extending the operator template space, but also developing mechanisms for handling noisy or partial tool outputs, and possibly adopting finer-grained credit assignment during training. We will include this point clearly in the revised Limitations section and consider it a core direction for future research.
>
> **$Q_2:$ The preference phase is fully offline.**
>
> $A_2:$ We thank the reviewer for this insightful observation. Our use of offline self-play preference optimization was intentional, as noted in Lines 202–211. We prioritized stability and reproducibility: online RL in this setting often suffers from sparse rewards and unstable convergence, which can obscure the true effect of dynamic planning.
>
> Despite being trained offline, the Designer shows strong zero-shot generalization. As shown in Table 1 and Table 3, it significantly improves performance on unseen domains like **SocialMaze** and **LiveBench**, demonstrating that DyFlow generalizes well without online adaptation.
>
> That said, we agree that continual improvement via online preference refinement or incremental self-play is a valuable future direction. We will note this explicitly in the revised *Limitations* section.
>
> **$Q_3:$ The paper proves DyFlow is “never worse” than a static policy by inclusion, but supplies no bound on performance loss due to planning error, nor on sample complexity of the KTO phase.**
>
> $A_3:$ We appreciate the reviewer’s suggestion. Our main focus is on demonstrating the practical effectiveness of dynamic workflow planning, rather than providing a tight theoretical bound. Nevertheless, as shown in Theorem 2 (Appendix B), DyFlow’s policy is guaranteed to be no worse than a static policy. A tighter, high-probability bound could indeed be derived by relating the per-step error $\epsilon_k$ to the probability of selecting a suboptimal subgraph (e.g., via Hoeffding-type concentration inequalities), but we leave such theoretical refinements to future work, as our contributions are primarily empirical.
>
> **$Q_4:$ Safety claims are anecdotal**
>
> $A_4:$ We thank the reviewer for highlighting this important point. All generated code was run in a controlled sandbox to ensure safety: (i) execution occurred in an isolated namespace, (ii) high-risk libraries (e.g., `os`, `sys`, `subprocess`) were blocked, and (iii) only a predefined `solve()` function was executed, with outputs and errors captured.
>
> Out of 164 HumanEval tasks, **163 ran successfully (99.4%)**. The one failure was due to infinite recursion, not unsafe or malicious behavior. We observed no hallucinated paths, unauthorized system calls, or prompt-injection attacks.
>
> We will clarify these quantitative results in the revision and note that broader stress testing (e.g., adversarial inputs, resource monitoring) is a valuable future direction.
>
> **$Q_5:$ Adding new capabilities still requires designing templates and writing prompt instructions**
>
> $A_5:$ We thank the reviewer for this constructive comment. We use dynamic operator to refer to the system's ability to adaptively compose and parameterize existing templates at runtime. This dynamic control over the workflow, as validated by the ablation in Table 4, is a core contribution. We agree that automatic operator induction is an important future direction and will clarify this in the Limitations section.
>
> **$Q_6:$ Critical settings (planner temperature 0.01, step cap $T_{\max}$) are fixed ad hoc. Sensitivity plots would clarify robustness**
>
> $A_6:$ We thank the reviewer for raising this point. The planner temperature was set to **0.01**. This choice follows common practice in reasoning agents and was adopted mainly to ensure deterministic decoding across different APIs. As some implementations internally treat 0 as a special case (allowing minimal stochasticity or fallback sampling in rare scenarios), using 0.01 enforces a strictly low-temperature regime while avoiding such implementation-dependent behavior.
>
> For the step cap $T_{\max}$, we chose **10** as a conservative upper bound. Among 164 HumanEval tasks, none required more than 7 stages (6 tasks used 6 steps, 2 used 7). Thus, $T_{\max} = 10$ safely covers all observed cases without incurring unnecessary computation.
>
> **$Q_7:$ How does performance degrade when the operator set is reduced or automatically expanded?**
>
> $A_7:$ We thank the reviewer for this valuable suggestion. To evaluate the contribution of each operator family, we ran an ablation using a simplified DyFlow with only five core templates (`REVIEW_SOLUTION`, `GENERATE_ANSWER`, `DECOMPOSE_PROBLEM`, `ORGANIZE_SOLUTION`, `TERMINATE`). We sampled 100 examples per benchmark and used the same Phi-4 executor.
>
> |Task|Full|Core|Δ|
> |----|----|----|--|
> |SocialMaze|17%|12%|−5%|
> |PubMedQA|71%|70%|−1%|
> |MATH|75%|71%|−4%|
> |LiveBench|48%|44%|−4%|
> |HumanEval|93%|90%|−3%|
>
> Removing higher-level operators consistently degrades performance, especially in tasks requiring flexible planning (e.g., **SocialMaze**). Automatic operator induction is beyond the scope of this work, and we will mention it as a key future direction in the revised *Limitations* section.
>
> **$Q_8:$ What fraction of tasks require a second or third re-planning step, and how often does re-planning fix an error vs cause thrashing?**
>
> $A_8:$ We thank the reviewer for this insightful question. In DyFlow, re-planning is triggered when the `REVIEW_SOLUTION` operator flags an intermediate error. To assess its impact, we manually reviewed 100 sampled trajectories on the most challenging benchmark, **SocialMaze**:
>
> - **Frequency:** 67 cases had one review, 25 had two, and 8 had three.
>
> - **Effectiveness:** 9 cases corrected an initial error; 2 introduced a new error after re-planning.
>
> Given SocialMaze’s difficulty (17% accuracy) and the limitations of the base Phi‑4 model, these results suggest that DyFlow’s re-planning mechanism is effective and introduces minimal thrashing. We will clarify this analysis in the final version.
>
> **$Q_9:$ Can the Designer trained under DyFlow self-play transfer to an executor of a different architecture with zero additional tuning?**
>
> $A_9:$ We thank the reviewer for this question. Yes, we have conducted such experiments, as shown in **Figure 4** and detailed in **Appendix D.5 (Table 10)**. The Designer trained under DyFlow self-play transfers to executors of different architectures (e.g., GPT-4.1-mini) with zero additional tuning and consistently improves performance over CoT baselines.
>
> This transferability is a direct consequence of DyFlow’s **decoupled architecture**: the Designer is responsible only for planning and data-flow orchestration, not for solving subproblems itself. Thus, it can treat any LLM as a black-box Executor, making zero-shot adaptation to new executors straightforward.
>
> **$Q_{10}:$ What are the worst observed failures and how are they detected or mitigated?**
>
> $A_{10}:$ We thank the reviewer for raising this important point. The worst failure we observed occurs when the Executor-generated code enters an **infinite or excessively long loop**. This was extremely rare (**1 out of 164 HumanEval tasks**) and was safely handled by our sandbox, which terminates execution upon timeouts.
>
> Moreover, we strictly prohibit high-risk libraries (`os`, `sys`, `subprocess`, etc.) and run all generated code in an isolated execution environment, ensuring that no OS-level crashes or security breaches can occur.
>
> **$Q_{11}:$ Could DyFlow be integrated with retrieval-augmented tool calls by treating search as an operator, and if so, how would the preference signal distinguish relevant from spurious retrievals?**
>
> $A_{11}:$ We thank the reviewer for this insightful comment. In our current setup, a **trajectory-level success/failure signal** is sufficient, as operators perform mostly deterministic reasoning and final outputs reliably reflect plan quality. This keeps preference optimization stable and low-variance.
>
> However, retrieval-augmented reasoning requires finer control. Trajectory-level feedback alone cannot distinguish useful from spurious retrievals. To support such cases, DyFlow would need **step-level quality signals** to reward relevant evidence and penalize noise—an important direction for future work.
>
> **$Q_{12}:$ Potential brittleness to summariser quality**
>
> $A_{12}:$ We appreciate the reviewer’s point. DyFlow’s summariser serves a narrowly scoped role: it condenses intermediate outputs into a concise state for the Designer, without performing complex reasoning. This summarization task is reliably handled by modern LLMs, and we observed very few cases of misleading or incomplete summaries. Even when minor omissions occur, DyFlow’s **multi-stage planning** allows later steps to detect inconsistencies and self-correct. In practice, such issues rarely impact the final outcome.
>
> ---
>
> Thank you again for your valuable time and suggestions and we would greatly appreciate your kind support during the discussion phase.

---

> > ### Comment · Reviewer_E5YQ · 2025-08-07
> >
> > Thank you for your detailed response. I have a better understanding to your work, and I would like to raise my scores.

---

> > > ### Author Response · Authors · 2025-08-07
> > >
> > > We deeply appreciate your thoughtful evaluation and your recognition of our contributions. Your detailed feedback and constructive suggestions have been instrumental in helping us refine our work, and we are truly thankful for your support during the review process.

---

### Decision · Program_Chairs · 2025-09-17

**Decision:**

Accept (poster)

**Comment:**

The submission proposed a workflow optimization solution. Basically, it includes two parts, 1). the designer to propose step wise plan, which is trained by SFT, self-play and KTO, and 2). the executor that can be any existing open or closed sourced LLM. The major contribution is to let the step-wise enviroment signal to guild the evolution of plans and thus effect the executor.
In summary, I do think it is an interesting work with solid contribution.
In the discussion period, all the reviewers agree to accept. As the agentic workflow automation and optimization are getting popular in the community, I thus champion the acceptance.